# UbiB proteins regulate cellular CoQ distribution in *Saccharomyces cerevisiae*

Zachary A. Kemmerer[1,2,9], Kyle P. Robinson[1,2,9], Jonathan M. Schmitz[1,2], Mateusz Manicki[3], Brett R. Paulson[4], Adam Jochem[1,2], Paul D. Hutchins[4], Joshua J. Coon [4,5,6] & David J. Pagliarini [1,2,3,7,8 ✉]

Beyond its role in mitochondrial bioenergetics, Coenzyme Q (CoQ, ubiquinone) serves as a key membrane-embedded antioxidant throughout the cell. However, how CoQ is mobilized from its site of synthesis on the inner mitochondrial membrane to other sites of action remains a longstanding mystery. Here, using a combination of *Saccharomyces cerevisiae* genetics, biochemical fractionation, and lipid profiling, we identify two highly conserved but poorly characterized mitochondrial proteins, Ypl109c (Cqd1) and Ylr253w (Cqd2), that reciprocally affect this process. Loss of Cqd1 skews cellular CoQ distribution away from mitochondria, resulting in markedly enhanced resistance to oxidative stress caused by exogenous polyunsaturated fatty acids, whereas loss of Cqd2 promotes the opposite effects. The activities of both proteins rely on their atypical kinase/ATPase domains, which they share with Coq8—an essential auxiliary protein for CoQ biosynthesis. Overall, our results reveal protein machinery central to CoQ trafficking in yeast and lend insights into the broader interplay between mitochondria and the rest of the cell.

[1] Morgridge Institute for Research, Madison, WI, USA. [2] Department of Biochemistry, University of Wisconsin-Madison, Madison, WI, USA. [3] Department of Cell Biology and Physiology, Washington University School of Medicine, St. Louis, MO, USA. [4] Department of Chemistry, University of Wisconsin-Madison, Madison, WI, USA. [5] Genome Center of Wisconsin, Madison, WI, USA. [6] Department of Biomolecular Chemistry, University of Wisconsin-Madison, Madison, WI, USA. [7] Department of Biochemistry and Molecular Biophysics, Washington University School of Medicine, St. Louis, MO, USA. [8] Department of Genetics, Washington University School of Medicine, St. Louis, MO, USA. [9]These authors contributed equally: Zachary A. Kemmerer, Kyle P. Robinson. ✉email: pagliarini@wustl.edu

CoQ is synthesized in mitochondria, where it functions as an essential cofactor in multiple processes including oxidative phosphorylation, fatty acid oxidation, and nucleotide biosynthesis[1–3]. CoQ is also present in membranes throughout the cell[4], suggesting that it has a more widespread cellular importance than is currently appreciated. Recently, one such role for extramitochondrial CoQ in mammalian cells was identified with the discovery that plasma membrane-localized FSP1 exhibits CoQ-dependent activity in mitigating ferroptosis[5,6], a form of regulated cell death caused by aberrant iron-dependent lipid peroxidation. To our knowledge, no proteins have yet been directly associated with cellular CoQ trafficking from mitochondria, but the extreme hydrophobicity of CoQ suggests that this process likely requires dedicated machinery.

Here, we demonstrate that two members of the poorly characterized UbiB family of atypical kinases/ATPases influence the cellular distribution of mitochondria-derived CoQ in the budding yeast *Saccharomyces cerevisiae*. We show that disruption of *CQD1* and *CQD2* diminishes and enhances the levels of mitochondrial CoQ, respectively, without altering total cellular CoQ abundance. Our findings help to define the functions of two mitochondrial proteins and advance our still nascent understanding of how CoQ is distributed throughout the cell.

## Results

**Extramitochondrial CoQ combats oxidative stress.** We sought to identify proteins related to CoQ trafficking by exploiting the extramitochondrial antioxidant role of $CoQ_6$—the major form of CoQ in *S. cerevisiae* (hereafter referred to as CoQ). *S. cerevisiae* lacking CoQ or phospholipid hydroperoxide glutathione peroxidases (PHGPx) are sensitive to the oxidative stress conferred by exogenous polyunsaturated fatty acids (PUFAs), such as α-linolenic acid (18:3)[7,8]. PUFAs undergo uncontrolled autoxidation reactions in the absence of these antioxidant factors, leading to the toxic accumulation of lipid peroxides and peroxyl radicals[7,8]. To force cells into relying more heavily on the antioxidant properties of CoQ, we deleted all three PHGPx genes in W303 *S. cerevisiae* Δ*gpx1*Δ*gpx2*Δ*gpx3* (hereafter referred to as Δ*gpx1/2/3*). We validated that this strain is sensitized to 18:3 treatment and demonstrated that this sensitivity is dampened when cellular CoQ levels are augmented through supplementation with the soluble CoQ precursor 4-hydroxybenzoate (4-HB) (Fig. 1a, b). Importantly, the CoQ analog decylubiquinone was markedly more effective at protecting against PUFA stress than its mitochondria-targeted counterpart, mitoquinone, suggesting that extramitochondrial CoQ is the predominant mediator of PUFA resistance (Fig. 1c). This is consistent with previous data showing that exogenous PUFAs are incorporated into endogenous membranes slowly[8] and, therefore, likely populate nonmitochondrial membranes first. Thus, we established a strain whose survival in the presence of PUFAs is especially dependent on extramitochondrial CoQ.

**Loss of Cqd1 confers PUFA resistance.** We reasoned that suppressor mutations that increase extramitochondrial CoQ levels would enhance PUFA resistance in the Δ*gpx1/2/3* strain, so we performed a forward-genetic suppressor screen (Fig. 2a). We randomly mutagenized this strain with ethyl methanesulfonate (EMS) and isolated colonies tolerant of 18:3 treatment. From ~20,000 unique mutant colonies, we obtained four hit strains with substantial PUFA resistance (Fig. 2b). We then performed whole-genome sequencing that revealed non-synonymous mutations in 442 unique genes across these four strains (Supplementary Data 1). These mutants were ranked using PROVEAN (Protein Variation Effect Analyzer), a software tool for predicting

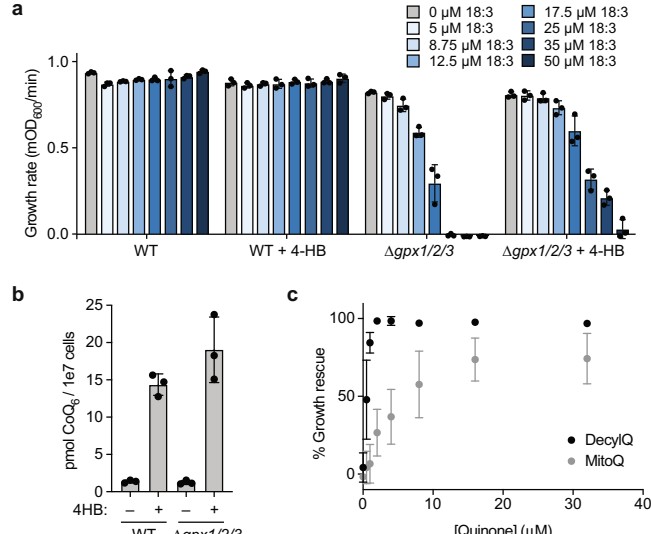

**Fig. 1 Extramitochondrial CoQ combats oxidative stress. a** Growth rate of wild type (WT) and Δ*gpx1/2/3* yeast in synthetic complete media minus para-aminobenzoate (*p*ABA⁻) containing 2% (w/v) glucose (mean ± SD, $n = 3$ independent samples) and the indicated additives. 4-HB, 4-hydroxybenzoate; 18:3, linolenic acid (PUFA). **b** Total CoQ from WT and Δ*gpx1/2/3* yeast described in (**a**) (mean ± SD, $n = 3$ independent samples). **c** Rescue assay under the conditions described in (**a**) comparing the ability of decylubiquinone (DecylQ) and mitoquinone (MitoQ) to restore growth of Δ*gpx1/2/3* yeast treated with 35 μM 18:3 (mean ± SD, $n = 3$ independent samples).

deleterious protein changes[9]. PROVEAN assigns a disruption score (D-Score) that reflects the likelihood that a given mutation is deleterious. In our collective dataset, 99 genes achieved a D-Score below the strict threshold of −4.1 (Fig. 2c; Supplementary Data 1). Given the overall limited overlap in hits between mutant strains, it is likely that our dataset includes multiple genes that contribute to an enhanced PUFA resistance phenotype.

We chose to focus on mitochondrial proteins for further examination since, to our knowledge, trafficking machinery at the site of CoQ synthesis in mitochondria has yet to be identified. Of the nine mitochondrial proteins harboring likely deleterious mutations, one, Ypl109c (renamed here as Cqd1, see below), is an uncharacterized protein that resides on the inner mitochondrial membrane (IMM), making it an attractive candidate for further study (Fig. 2c; Supplementary Fig. 1a). Moreover, Cqd1 possesses the same UbiB family atypical kinase/ATPase domain as Coq8, an essential protein for CoQ synthesis that resides on the matrix face of the IMM[10–13]. Our recent work suggests that Coq8 ATPase activity may be coupled to the extraction of hydrophobic CoQ precursors from the IMM for subsequent processing by membrane-associated matrix enzymes[14]. Cqd1 resides on the opposite side of the IMM, facing the intermembrane space[11,15] (Supplementary Fig. 1b), physically separated from the other CoQ-related enzymes but still positioned for direct access to membrane-embedded CoQ precursors and mature CoQ. Furthermore, a recent study reported that haploinsufficiency of human *CQD1* ortholog *ADCK2* led to aberrant mitochondrial lipid oxidation and myopathy associated with $CoQ_{10}$ deficiency[16].

In our screen, each resistant strain (mutA-D) possesses more than 100 protein-coding mutations, a combination of which likely contributes to the PUFA resistance phenotype. Mutant C (mutC) contains an early stop codon in *CQD1* (Fig. 2c, Supplementary Fig. 1c). To test whether this *CQD1* mutation is important for mutC's phenotype, we reintroduced WT *CQD1* into this strain under its endogenous promoter. Indeed, this reintroduction re-

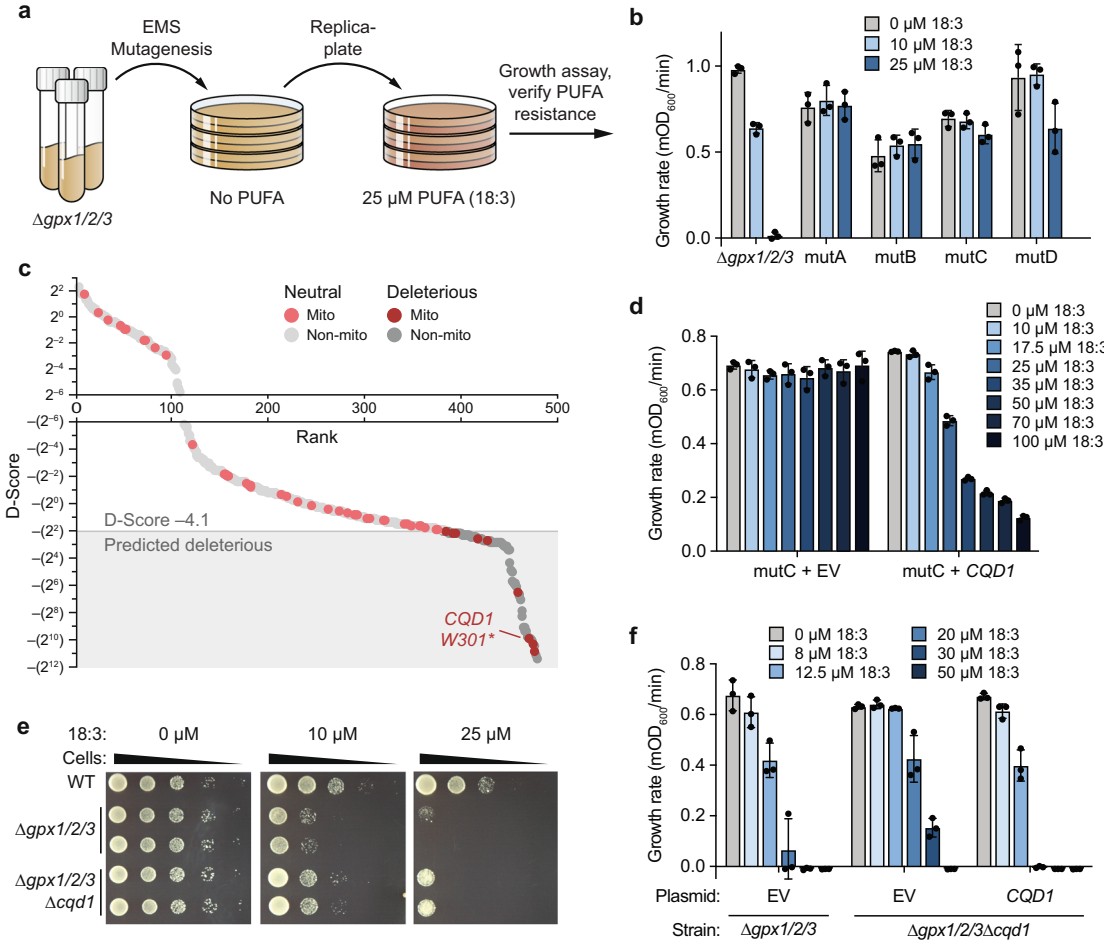

**Fig. 2 Genome-wide screen for CoQ trafficking genes identifies uncharacterized UbiB protein Cqd1. a** Schematic of forward-genetic yeast screen for genes involved in CoQ trafficking. **b** Growth rates of $\Delta gpx1/2/3$ and four mutant strains resistant to 18:3 treatment (mutA-D). Yeast were assayed in $p$ABA⁻ media containing 2% (w/v) glucose with 0−25 μM 18:3 (mean ± SD, $n = 3$ independent samples). **c** Mutant strains mutA-D were submitted for whole-genome sequencing to identify non-synonymous mutations (total = 442). Mutations were analyzed with PROVEAN to filter for likely deleterious changes (D-score ≤ −4.1, shaded box). Gray, all genes; red, mitochondrial genes. Light, predicted neutral; dark, predicted deleterious. **d** Growth rate of mutC yeast expressing empty vector (EV) or endogenous *CQD1* (mean ± SD, $n = 3$ independent samples). Yeast were assayed under the conditions described in (**b**) with 0−100 μM 18:3. **e** Drop assay of WT, $\Delta gpx1/2/3$, and $\Delta gpx1/2/3\Delta cqd1$ yeast grown for 3 days on solid $p$ABA⁻ medium containing 2% (w/v) glucose, 0.5% (w/v) ethanol (EtOH), and 0-25 μM 18:3. A representative drop assay from three independent experiments is shown. **f** Growth rates of $\Delta gpx1/2/3$ and $\Delta gpx1/2/3\Delta cqd1$ yeast expressing EV or endogenous *CQD1* (mean ± SD, $n = 3$ independent samples). Yeast were assayed under the conditions described in (**b**) with 0−50 μM 18:3. Source data for panel (**c**) is provided as a Source Data file.

conferred PUFA sensitivity (Fig. 2d). Furthermore, deletion of *CQD1* in the parent $\Delta gpx1/2/3$ strain, which lacks all other mutC mutations, was sufficient to enhance PUFA resistance (Fig. 2e, f). We also confirmed that deletion of *CQD1* had no effect in background strains lacking CoQ ($\Delta coq2$ and $\Delta gpx1/2/3\Delta coq2$), establishing that this PUFA-resistant phenotype is CoQ-dependent (Supplementary Fig. 1d, e). Collectively, these data demonstrate that disruption of *CQD1* is at least partially causative for mutC's PUFA-resistant phenotype.

**Cqd1 affects CoQ distribution.** Our results above suggest that loss of *CQD1* confers cellular resistance to PUFA-mediated oxidative stress by increasing extramitochondrial CoQ. We reasoned that this was likely rooted either in a general increase in CoQ production or in its redistribution. To test these models, we first measured total levels of CoQ and its early mitochondrial precursor polyprenyl-4-hydroxybenzoate (PPHB) in cells lacking *CQD1* or control genes (Fig. 3a−c). As expected, disruption of *HFD1*, which encodes the enzyme that produces the soluble CoQ

precursor 4-HB[17,18], led to a loss of CoQ and PPHB, while disruption of *COQ8* caused complete loss of CoQ with the expected buildup of the PPHB precursor. However, we found no significant change in CoQ or PPHB levels in the $\Delta cqd1$ strain, demonstrating that Cqd1 is essential neither for CoQ biosynthesis nor the import of CoQ precursors under the conditions of our analyses.

To next examine CoQ distribution, we fractionated yeast and measured CoQ levels (Fig. 3d; Supplementary Fig. 2a). We observed that $\Delta cqd1$ yeast had a significant increase in CoQ from the non-mitochondrial (NM) fraction, consisting of organelles and membranes that do not pellet with mitochondria, and a corresponding significant decrease in mitochondrial (M) CoQ. Deletion of the tricarboxylic acid (TCA) cycle enzyme Kgd1 had no effect on relative CoQ levels (Fig. 3d) despite causing a deficiency in respiratory growth (Fig. 3e), indicating that general mitochondrial dysfunction does not perturb CoQ distribution. The increased extramitochondrial CoQ in $\Delta cqd1$ yeast is consistent with the observation that deleting *CQD1* increases PUFA resistance (Fig. 2e, f).

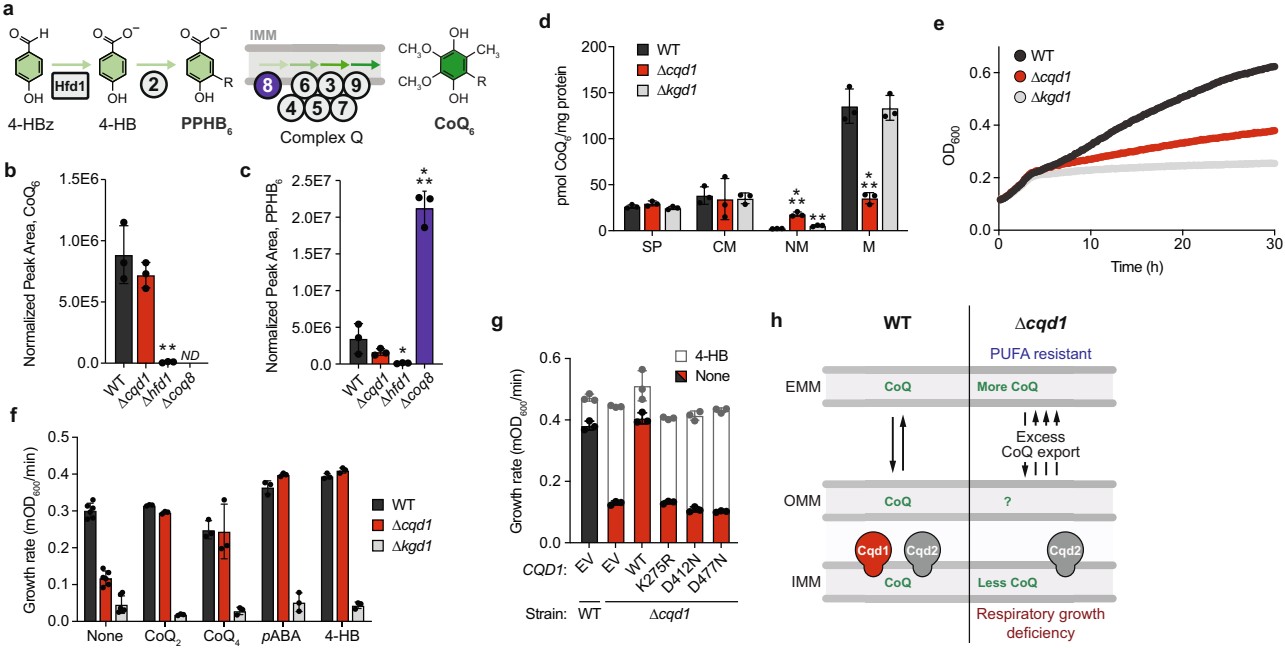

**Fig. 3 Cqd1 influences cellular CoQ distribution. a** Schematic of CoQ biosynthesis pathway. Polyprenyl hydroxybenzoate (PPHB) is an early precursor that undergoes a series of head group modifications by IMM-associated Coq enzymes (complex Q) to produce CoQ. Hfd1 is essential for PPHB synthesis, while Coq8 is required for the production of CoQ. **b** Total CoQ from WT, $\Delta cqd1$, $\Delta hfd1$, and $\Delta coq8$ yeast (**$p = 0.0030$ WT vs $\Delta hfd1$; mean ± SD, $n = 3$ independent samples); not detected ND. **c** Total polyprenyl-4-hydroxybenzoate (PPHB) from WT, $\Delta cqd1$, $\Delta hfd1$, and $\Delta coq8$ yeast (*$p = 0.0471$ WT vs $\Delta hfd1$, ***$p = 0.0006$ WT vs $\Delta coq8$; mean ± SD, $n = 3$ independent samples). **d** CoQ from subcellular fractions derived from WT, $\Delta cqd1$, and $\Delta kgd1$ yeast (***$p = 0.0004$ WT NM vs $\Delta cqd1$ NM, **$p = 0.0029$ WT NM vs $\Delta kgd1$ NM, ***$p = 0.0009$ WT M vs $\Delta cqd1$ M; mean ± SD, $n = 3$ independent samples). SP, spheroplast; CM, crude mitochondria; NM, non-mitochondrial fraction; M, enriched mitochondria. **e** Growth assay of WT, $\Delta cqd1$, and $\Delta kgd1$ yeast in $p$ABA$^-$ media containing 0.1% (w/v) glucose and 3% (w/v) glycerol (mean, $n = 6$ independent samples). Yeast enter the respiratory phase of growth after ~4 h in this growth condition. **f** Growth rate of WT, $\Delta cqd1$, and $\Delta kgd1$ yeast assayed under conditions described in (**d**) (mean ± SD; none $n = 6$ independent samples, all others $n = 3$). Yeast were grown in the presence and absence of 100 μM CoQ analogs (CoQ$_2$, CoQ$_4$) and 1 μM CoQ precursors ($p$ABA, 4-HB). **g** Growth rate of WT and $\Delta cqd1$ yeast transformed with the indicated plasmids (EV, $CQD1$ or $CQD1$, point mutants) and grown in Ura$^-$, $p$ABA$^-$ media containing 0.1% (w/v) glucose and 3% (w/v) glycerol (mean ± SD, $n = 3$ independent samples). Yeast were treated with 0 (colored bars) or 1 μM 4-HB (white bars, superimposed) to determine rescue of respiratory growth. **h** Model for Cqd1's putative role in cellular CoQ distribution. OMM, outer mitochondrial membrane; IMM, inner mitochondrial membrane; EMM, extramitochondrial membranes. **b–d** Significance calculated by an unpaired, two-tailed Student's $t$-test.

To our knowledge, this is the first example of a genetic disruption leading to altered cellular distribution of endogenous CoQ, hence our renaming of this gene <u>Co</u>Q <u>D</u>istribution <u>1</u> (*CQD1*). To further validate this finding, we examined growth in glycerol, a non-fermentable carbon source, which requires an intact mitochondrial electron transport chain. We reasoned that a decrease in mitochondrial CoQ would disrupt respiratory growth in media depleted of CoQ precursors. Indeed, deletion of *CQD1* reduced respiratory growth rate in this medium appreciably (Fig. 3e). To confirm that this defect is caused by CoQ depletion, we rescued growth with CoQ of different isoprene tail lengths (CoQ$_2$ and CoQ$_4$) and with CoQ precursors, which are more readily delivered due to their solubility (Fig. 3f). Endogenous expression of *CQD1* rescued respiratory growth without affecting total CoQ levels (Fig. 3g, Supplementary Fig. 2b), further supporting the hypothesis that CoQ distribution, not biosynthesis, is perturbed in $\Delta cqd1$ yeast.

We next sought to begin understanding how Cqd1 functions in CoQ distribution. Our recent work on Cqd1's UbiB homolog COQ8 (yeast Coq8 and human/mouse COQ8A) revealed that it possesses an atypical protein kinase-like (PKL) fold that endows ATPase activity but occludes larger proteinaceous substrates from entering the active site[13,19] (Supplementary Fig. 2c–e). Unlike COQ8, Cqd1 is recalcitrant to recombinant protein purification; therefore, in lieu of direct in vitro activity assays, we examined the

ability of Cqd1 point mutants to rescue the respiratory growth defect of $\Delta cqd1$ yeast. Similar to Coq8[13,14,19], the ability of Cqd1 to rescue the $\Delta cqd1$ respiratory growth deficiency depended on core protein kinase-like (PKL) family residues[20] required for phosphoryl transfer (Fig. 3g) and on quintessential UbiB motif residues (Supplementary Fig. 2e–h). Further biochemical work is required to prove Cqd1's enzymatic activity; however, these data support a model whereby Cqd1's ability to promote CoQ distribution relies on atypical kinase/ATPase activity (Fig. 3h).

**Cqd2 counteracts Cqd1 function.** Beyond Coq8 and Cqd1, the *S. cerevisiae* genome encodes just one other member of the UbiB family—Ylr253w (aka Mcp2, and renamed here Cqd2). Cqd2 is also poorly characterized and resides in the same location as Cqd1, on the outer face of the IMM[11,15,21] (Supplementary Fig. 1b). Previous studies have identified genetic and physical interactions connecting Cqd2 to mitochondrial lipid homeostasis, but not to a specific pathway[21–23]. Given the similarity between these three proteins (Supplementary Fig. 2d, e), we anticipated that Cqd2 might also be connected to CoQ biology.

To test this hypothesis, we disrupted *CQD2* in $\Delta gpx1/2/3$ yeast and subjected this strain to PUFA-mediated stress. Surprisingly, $\Delta gpx1/2/3\Delta cqd2$ yeast exhibited an enhanced sensitivity to PUFA treatment—the opposite phenotype to that of $\Delta gpx1/2/3\Delta cqd1$

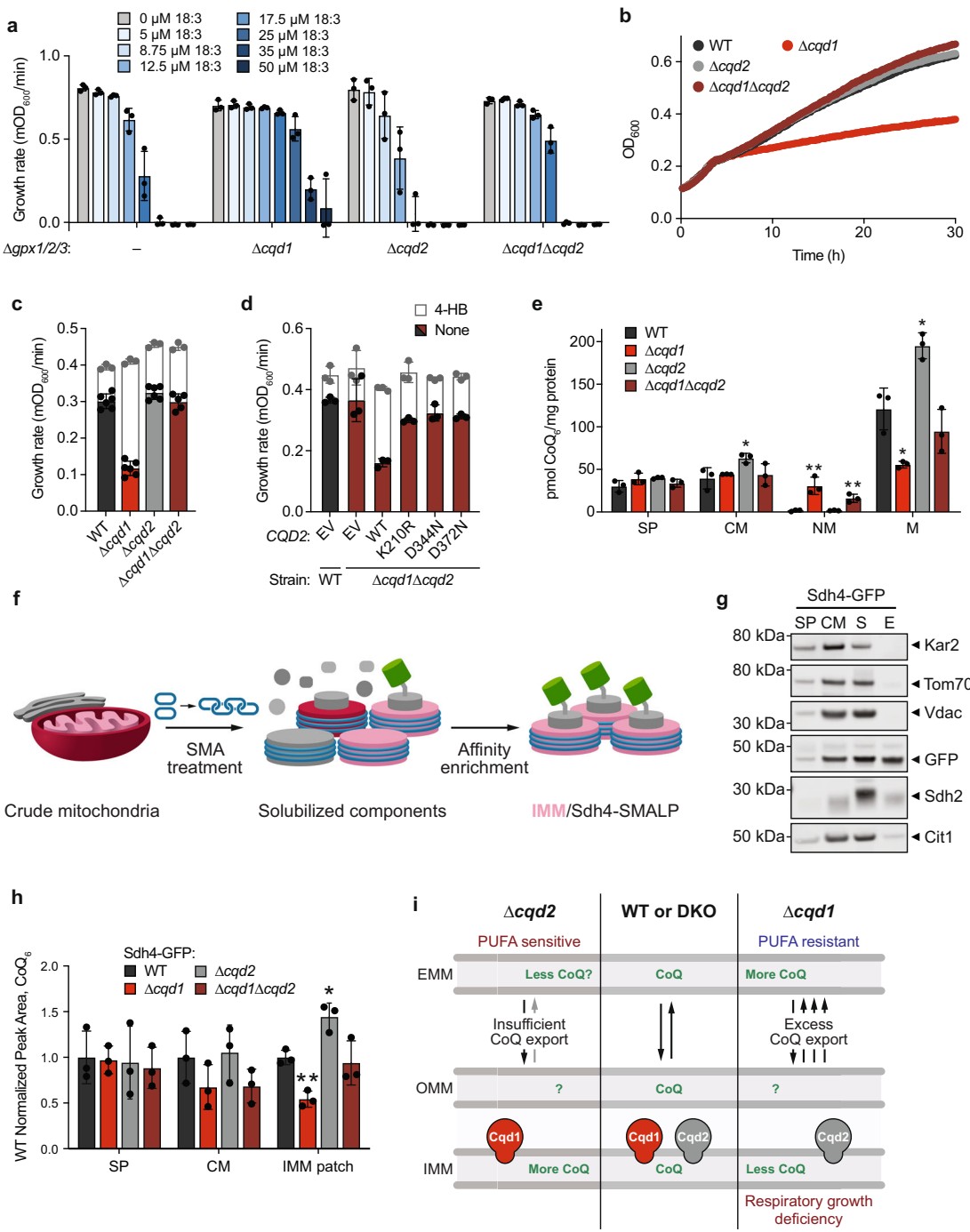

(Fig. 4a; Supplementary Fig. 3a). This phenotype is also CoQ-dependent, as deletion of *CQD2* likewise had no effect in background strains lacking CoQ (Supplementary Fig. 1d, e). Furthermore, Δgpx1/2/3Δcqd1Δcqd2 yeast phenocopied the parental (Δgpx1/2/3) strain (Fig. 4a; Supplementary Fig. 3a). Under respiratory conditions, Δcqd2 yeast exhibited no detectable change in growth. However, deleting *CQD2* from Δcqd1 yeast (Δcqd1Δcqd2) restored this strain's impaired respiratory growth rate to WT levels (Fig. 4b, c). Conversely, reintroduction of *CQD2* into the Δcqd1Δcqd2 strain recapitulated the respiratory growth deficiency of Δcqd1 (Fig. 4d). Total cellular CoQ levels remained unchanged (Supplementary Fig. 3b), again suggesting these CoQ-related phenotypes are unrelated to CoQ biosynthesis. Similar to Cqd1 (Fig. 3g), Cqd2 function was dependent on intact canonical PKL and UbiB-specific residues (Fig. 4d, Supplementary Fig. 3c–e),

suggesting that all three UbiB family proteins in yeast are active phosphoryl transfer enzymes. Consistent with these results, subcellular fractionation revealed significantly increased CoQ levels in the pure mitochondrial fraction from Δcqd2 yeast (Fig. 4e; Supplementary Fig. 3f). Furthermore, the Δcqd1Δcqd2 strain possessed mitochondrial and non-mitochondrial CoQ levels between those of the Δcqd1 and Δcqd2 strains (Fig. 4e). However, our fractionation approach, which prioritizes high purity over yield, only detected very low levels of CoQ in the WT and Δcqd2 non-mitochondrial samples; therefore, a quantifiable loss of CoQ in this fraction for the Δcqd2 was not detectable (Fig. 4e).

The analyses above, coupled with the submitochondrial location of Cqd1 and Cqd2, suggest a model whereby these enzymes may reciprocally regulate the amount of CoQ within the IMM. To test this directly, we used the amphipathic polymer

**Fig. 4 Cqd2 function opposes Cqd1 control of CoQ distribution. a** Growth rate of Δ*gpx1/2/3* and the described yeast strains in *p*ABA⁻ media containing 2% (w/v) glucose and the indicated additives (mean ± SD, *n* = 3 independent samples). **b** Growth assay of WT, Δ*cqd1*, Δ*cqd2*, and Δ*cqd1*Δ*cqd2* yeast in *p*ABA⁻ media containing 0.1% (w/v) glucose and 3% (w/v) glycerol (mean ± SD, *n* = 6 independent samples). **c** Growth rate of yeast strains in b treated with 0 (colored bars) or 1 μM 4-HB (white bars, superimposed) (mean ± SD; 0 μM 4-HB *n* = 6 independent samples, 1 μM 4-HB *n* = 3). **d** Growth rate of WT and Δ*cqd1*Δ*cqd2* yeast transformed with the indicated plasmids (EV, *CQD2*, or *CQD2* point mutants) and grown in Ura⁻, *p*ABA⁻ media containing 0.1% (w/v) glucose and 3% (w/v) glycerol (mean ± SD, *n* = 3 independent samples). Yeast were treated with 0 (colored bars) or 1 μM 4-HB (white bars, superimposed) to determine recapitulation of respiratory growth defect. **e** CoQ from subcellular fractions derived from WT, Δ*cqd1*, Δ*cqd2*, and Δ*cqd1*Δ*cqd2* yeast (**p* = 0.0392 WT CM vs Δ*cqd2* CM, ***p* = 0.0081 WT NM vs Δ*cqd1* NM, ***p* = 0.0075 WT NM vs Δ*cqd1*Δ*cqd2* NM, **p* = 0.0105 WT M vs Δ*cqd1* M, **p* = 0.0112 WT M vs Δ*cqd2* M; mean ± SD, *n* = 3 independent samples). SP, spheroplast; CM, crude mitochondria; NM, non-mitochondrial fraction; M, enriched mitochondria. **f** Schematic of Sdh4-GFP styrene-maleic acid (SMA) lipid particle (SMALP) isolation. **g** Western blot to assess the purity of SMALP isolation samples from endogenously tagged Sdh4-GFP yeast. SP, spheroplast; CM, crude mitochondria; S, soluble; E, elution (or IMM patch). Kar2, endoplasmic reticulum; Tom70, outer mitochondrial membrane (OMM); Vdac, OMM; Sdh4-GFP, SMALP target/IMM; Sdh2, IMM; Cit1, mitochondrial matrix. A representative western blot from three independent experiments. **h** CoQ from subcellular fractions derived from SMALP isolation described in (**f**) for the indicated strains (***p* = 0.0026 WT IMM patch vs Δ*cqd1* IMM patch, **p* = 0.0114 WT IMM patch vs Δ*cqd2* IMM patch; mean ± SD, *n* = 3). **i** Summary model depicting opposing roles for yeast UbiB family proteins in the cellular distribution of CoQ. OMM, outer mitochondrial membrane; IMM, inner mitochondrial membrane; EMM, extramitochondrial membranes. **e**, **h** Significance calculated by an unpaired, two-tailed Student's *t*-test.

styrene-maleic acid (SMA) to solubilize integral membrane proteins into detergent-free SMA lipid particles (SMALPs)[24] from yeast harboring an endogenously-tagged subunit of mitochondrial complex II (Sdh4-GFP). Recent work validated this approach as an effective method for parsing mitochondrial membranes and isolating pure IMM content[25]. We reasoned that purifying lipid patches containing Sdh4, which directly interacts with CoQ to facilitate succinate dehydrogenase (Complex II) activity[26], would yield a suitable lipid microenvironment to measure IMM-localized CoQ. After solubilization (Supplementary Fig. 3g), we isolated native IMM patches that possessed Sdh4-GFP using a recombinantly purified His-tagged GFP nanobody (Fig. 4f; Supplementary Fig. 3h, i). We show that purified Sdh4-GFP IMM patches are largely void of extramitochondrial and outer mitochondrial membrane (OMM) protein contamination (Fig. 4g), making this a reliable sample for assessing IMM CoQ abundance.

We generated a panel of deletion strains in the Sdh4-GFP background to investigate how the loss of Cqd1 and Cqd2 impacts CoQ abundance in this IMM microenvironment. These yeast strains exhibited the same respiratory phenotypes as the W303 background strains and had similar levels of whole-cell CoQ (Supplementary Fig. 3j, k). After solubilization and affinity enrichment (Supplementary Fig. 3l, m), Sdh4-GFP IMM patch lipids were extracted for targeted CoQ measurements. Consistent with our respiratory growth and fractionation observations, Δ*cqd1* yeast had significantly lower levels of IMM patch CoQ. Conversely, the Δ*cqd2* yeast had elevated IMM patch CoQ, while Δ*cqd1*Δ*cqd2* yeast had levels similar to the parental strain (Fig. 4h). These data provide additional evidence of protein-dependent changes in CoQ distribution, corroborating our phenotypic observations. Taken together, our results suggest that Cqd1 and Cqd2 reciprocally regulate the levels of IMM CoQ and support a model wherein proper cellular CoQ distribution is dependent on the balance of their activities (Fig. 4i).

Once extracted from the IMM, we expect that subsequent steps would be required to deliver CoQ to extramitochondrial membranes (EMM). The multimeric ER-mitochondrial encounter structure (ERMES) and mitochondrial contact site and cristae organizing system (MICOS) complexes facilitate interorganellar lipid and metabolite transfer[27,28]. Recent work has revealed that CoQ biosynthetic machinery and MICOS subcomplexes often colocalize with ERMES[29–31], suggesting that these sites could serve as conduits for CoQ transport. To investigate the role of ERMES and MICOS in intramitochondrial CoQ trafficking, we disrupted a key subunit from each protein complex—*MDM34* or *MIC60*, respectively. Disruption of ERMES or MICOS had no effect on

PUFA resistance observed after subsequent deletion of *CQD1*, suggesting neither ERMES nor MICOS is required for the increased CoQ export in Δ*cqd1* strains (Supplementary Fig. 4a, b).

To explore this approach more thoroughly, we also deleted *MCP1*, a subunit of the vacuolar and mitochondria patch (vCLAMP) complex[32], as well as genes associated with inter-organellar (*LTC1*)[33] and intramitochondrial (*MDM31*)[34,35] lipid homeostasis. Remarkably, none of these deletions blocked the increased PUFA resistance upon *CQD1* knockout, suggesting that these genes are also not required for mitochondrial CoQ export (Supplementary Fig. 4c, d). Thus, *CQD1* and *CQD2* are the sole genes currently associated with the redistribution of mitochondrial CoQ.

## Discussion
Our work demonstrates that two previously uncharacterized UbiB family proteins influence the cellular distribution of mitochondria-derived CoQ. To our knowledge, Cqd1 and Cqd2 are the first proteins implicated in this process, which is essential for providing membranes throughout the cell with the CoQ necessary for enzymatic reactions and antioxidant defense. Further efforts are needed to establish how these proteins support CoQ distribution mechanistically; however, their similarity to Coq8 and the requirement for canonical PKL residues in their active sites suggests that Cqd1 and Cqd2 may couple ATPase activity to the selective extraction/deposition of CoQ from/to the IMM.

Our investigations here focused on CoQ; however, it is possible that Cqd1 and Cqd2 (aka Mcp2) influence lipid transport and homeostasis more broadly. Previous work has identified an array of genetic interactions for Cqd1 and Cqd2 with lipid biosynthesis and homeostasis genes[23,36]. Moreover, Cqd2 was previously identified as a high-copy suppressor of a growth defect caused by loss of the ERMES subunit Mdm10[21]. More recently, three conserved Cqd2 active site residues were shown to mitigate rescue of Δ*mdm10* yeast growth[22], results that we confirm (Cqd2 K210R) and expand upon with six additional residue mutations.

Interestingly, mitochondrial CoQ export still occurs in the absence of Cqd1/2, indicating that additional factors can participate in this process. This observation is consistent with multiple other recent studies demonstrating that various aspects of phospholipid transport are highly redundant in yeast[28]. For example, ERMES and vCLAMP appear to have overlapping functions while normally operating under different growth conditions[37,38]. In the absence of Cqd1 and Cqd2, CoQ transport between the OMM and IMM might be achieved by a combination of MICOS and other lipid-binding proteins. Although our data demonstrate that

MICOS disruption is insufficient to thwart the PUFA resistance mediated by disruption of *CQD1*, MICOS alone is often not sufficient to facilitate lipid movement between these membranes, which instead relies on dedicated phospholipid trafficking proteins[39]. COQ9 is a lipid-binding protein that likely delivers CoQ precursors to matrix enzymes[40], suggesting that other lipid-binding proteins may indeed exist to shuttle CoQ. Our genetic screen has nominated several extramitochondrial and cytosolic proteins as promising leads for these processes, and validating additional causative mutations for mutants A−D will be a focal point of our future work. Moreover, our discovery of Cqd1 and Cqd2 should accelerate the discovery of other proteins in these pathways (e.g., by performing similar screens in a Δ*cqd1*Δ*cqd2* background). Of note, although Gpx1-3 are primarily cytosolic[41,42], they have been localized to mitochondria[11,15,43]. Thus, our screen may also be equipped to identify genes that protect against loss of mitochondrial-based Gpx defenses.

Finally, UbiB family proteins are found across all domains of life[44]. UbiB homologs in plants (termed ABC1K proteins) are abundant, with 17 found in *Arabidopsis*[45]. Many of these ABC1K proteins are localized to plastoglobules—plastid-localized lipoprotein particles that contain various lipid-derived metabolites—and recent work suggests that ABC1K1 and ABC1K3 may affect the mobility and exchange of their subcellular plastoquinone-9 pools[46], suggesting UbiB proteins might function in quinone distribution across species. In humans, five UbiB proteins have been identified, ADCK1-5. While COQ8A (ADCK3) and COQ8B (ADCK4) have established roles in CoQ biosynthesis and human disease[13,47,48], the biological roles of other ADCK proteins remain elusive. Genome-wide knockdown studies have implicated these uncharacterized *ADCK* genes in several cancer disease states[49−52]. As novel targets for human disease intervention, it will be important to determine if functional conservation exists between Cqd1 and Cqd2 and their putative human orthologs, ADCK2 and ADCK1/5, respectively. Recently, a crucial new role for extramitochondrial CoQ was identified in mitigating ferroptosis, a type of cell death stemming from a buildup of toxic lipid peroxides, suggesting that manipulating CoQ distribution could provide therapeutic benefits[5,6]. Notably, we have developed small-molecule modulators for Coq8[14] and COQ8A[53], indicating that UbiB proteins are promising druggable targets.

Collectively, our work to de-orphanize these poorly characterized mitochondrial proteins represents the first step in addressing enduring questions regarding endogenous cellular CoQ distribution and unlocking the therapeutic potential of manipulating this pathway.

## Methods

**Yeast strains and cultures.** Unless otherwise described, *Saccharomyces cerevisiae* haploid W303 (MATa his3 leu2 met15 trp1 ura3) yeast were used. For SMA-derived lipid nanodisc work, endogenous GFP-tagged BY4741 (MATa his3Δ1 leu2Δ0 met15Δ0 ura3Δ0) yeast strains[54] were used. Yeast deletion strains were generated using standard homologous recombination or CRISPR-mediated methods (all primers used in this study are detailed in Supplementary Data 2). For homologous recombination, open reading frames were replaced with the KanMX6, HygMX6, or NatMX6 cassette as previously described[55]. Cassette insertion was confirmed by a PCR assay and DNA sequencing. CRISPR-mediated deletions were performed as described in[56]. 20-mer guide sequences were designed with the ATUM CRISPR gRNA design tool (https://www.atum.bio/eCommerce/cas9/input) and cloned into pRCC-K, and 500 ng of the guide-inserted pRCC-K was used per yeast transformation. Donor DNA was 300 pmol of an 80-nt Ultramer consisting of 40 bp upstream and 40 bp downstream of the ORF (for scarless deletions) or ~6 μg of PCR-amplified Longtine cassette with flanking homology 40 bp upstream and 40 bp downstream of the ORF (for cassette-replacement deletions).

Synthetic complete (and dropout) media contained drop-out mix (US Biological), yeast nitrogen base (with ammonium sulfate and without amino acids) (US Biological), and the indicated carbon source. *p*ABA⁻ (and dropout) media contained Complete Supplement Mixture (Formedium), Yeast Nitrogen Base without amino acids and without *p*ABA (Formedium), and the indicated carbon source. All media were sterilized by filtration (0.22 μm pore size).

**Yeast growth assay and drop assay**

*PUFA growth assays.* To assay yeast growth in liquid media, individual colonies were used to inoculate synthetic complete (or synthetic complete dropout) media (2% glucose, w/v) starter cultures, which were incubated overnight (30 °C, 230 rpm). Yeast were diluted to $1.1 \times 10^6$ cells/mL in *p*ABA⁻ (or *p*ABA⁻ dropout) media (2% glucose, w/v) and incubated until early phase (30 °C, 7−8 h, 230 rpm). Yeast were swapped into fresh *p*ABA⁻ media (2% glucose, w/v) at an initial density of $5 \times 10^6$ cells/mL with indicated additives. The cultures were incubated (30 °C, 1140 rpm) in an Epoch2™ plate reader (BioTek®) in a sterile 96 well polystyrene round bottom microwell plate (Thermo) with a Breathe-Easy® cover seal (Diversified Biotech). Optical density readings ($A_{600}$) were obtained every 10 min, and growth rates were calculated with Gen5 v3.02.2 software (BioTek®), excluding timepoints from the stationary phase.

*Respiratory growth assays.* Individual colonies of *S. cerevisiae* were used to inoculate synthetic complete media (2% glucose, w/v) starter cultures, which were incubated overnight (30 °C, 230 rpm). For transformed yeast strains, the corresponding Ura⁻ media was used. Yeast were diluted to $1 \times 10^6$–$1.33 \times 10^6$ cells/mL in *p*ABA⁻ media (2% glucose, w/v) and incubated until early log phase (30 °C, 7−8 h, 230 rpm). Yeast were swapped into *p*ABA⁻ media with glucose (0.1%, w/v) and glycerol (3%, w/v) at an initial density of $5 \times 10^6$ cells/mL with indicated additives. The cultures were incubated (30 °C, 1140 rpm) in an Epoch2 plate reader (BioTek) in a sterile 96 well polystyrene round bottom microwell plate (Thermo) with a Breathe-Easy cover seal (Diversified Biotech). Optical density readings ($A_{600}$) were obtained every 10 min, and growth rates were calculated with Gen5 v3.02.2 software (BioTek), excluding time points before the diauxic shift and during stationary phase growth.

*Drop assays.* Individual colonies of yeast were used to inoculate *p*ABA-limited media (2% w/v glucose, 100 nM *p*ABA) starter cultures, which were incubated overnight (30 °C, 230 rpm). Cells were spun down (21,000 × *g*, 2 min) and resuspended in water. Serial dilutions of yeast ($10^5$, $10^4$, $10^3$, $10^2$, or 10 cells) were dropped onto *p*ABA⁻ media (2% glucose and 1% EtOH, w/v) agar plates with indicated additives and incubated (30 °C, 2−3 d).

**Forward-genetic screen.** Individual colonies of Δ*gpx1/2/3* yeast were used to inoculate YEPD starter cultures, which were incubated overnight. $1.0 \times 10^8$ cells were pelleted, washed once with sterile water, and resuspended in 2.5 mL of 100 mM sodium phosphate buffer, pH 7.0. Ethyl methanesulfonate (EMS) (80 μL) was added, and cells were incubated (90 min, 30 °C, 230 rpm). Cells were washed thrice with sodium thiosulfate (5% w/v) to inactivate EMS. Cells were resuspended in water, and $1.0 \times 10^4$ cells were plated on *p*ABA-limited (2% w/v glucose, 100 nM *p*ABA) agar plates. After 3 days, cells were replica-plated onto *p*ABA⁻ (2% glucose, w/v) plates with 0 μM or 25 μM α-linolenic acid (C18:3, Sigma). Colonies that grew on 25 μM linolenic acid were picked into YEPD overnight cultures and struck on YEPD plates, and PUFA resistance phenotypes were confirmed with plate reader growth assays. For mutant strains that grew in the presence of 25 μM linolenic acid, genomic DNA was isolated with the MasterPure™ Yeast DNA Purification Kit (Lucigen) and submitted to GENEWIZ for whole-genome sequencing. *S. cerevisiae* genome assembly and variation calling were performed with SeqMan NGen 14 and ArrayStar 14 (DNASTAR Lasergene suite). Variant D-Score predictions were obtained using the PROVEAN v1.1.3 web server (http://provean.jcvi.org/seq_submit.php).

**Plasmid cloning.** Expression plasmids were cloned with standard restriction enzyme cloning methods. ORF-specific primers (Supplementary Data 2) were used to amplify Cqd1 (Ypl109c) and Cqd2 (Ylr253w) from W303 yeast genomic DNA. Amplicons were treated with DpnI to degrade genomic DNA and ligated into the digested p416 GPD plasmid (Addgene). Cloning products were then transformed into *E. coli* 10G chemically competent cells (Lucigen). Plasmids were isolated from transformants and Sanger sequencing was used to identify those containing the correct insertion.

Constructs containing Cqd1 and Cqd2 were digested with SalI and BamHI or HindIII to liberate the GPD promoter. Digested backbones were then combined with amplified endogenous promoter regions (1000 bases upstream for Cqd1, 500 bases upstream for Cqd2) and ligated to generate endogenous promoter vectors for Cqd1 and Cqd2.

**Site-directed mutagenesis.** Point mutants were constructed as described in the Q5® Site-Directed Mutagenesis Kit (New England Biolabs) and were confirmed via Sanger sequencing. Yeast were transformed as previously described[57] with plasmids encoding Cqd1 and Cqd2 variants with their endogenous promoters and grown on uracil drop-out (Ura⁻) synthetic media plates containing glucose (2%, w/v).

**Homology model generation.** Amino acid sequences of Cqd1 and Cqd2 were threaded through COQ8A apo crystal structure (PDB:4PED) via the online I-TASSER webserver[58]. Superimposed homology models were visualized in the PyMOL Molecular Graphics System (Version 2.0, Schrödinger, LLC). Color

schemes depicting protein domain organization were chosen according to the previous work[19].

**Subcellular fractionation**. Individual colonies of S. cerevisiae were used to inoculate synthetic complete media (2% glucose, w/v) starter cultures and were incubated for 14−16 h (30 °C, 230 rpm). Yeast were diluted to $5 \times 10^6$ cells/mL in 50 mL pABA⁻ media (2% glucose, w/v) and incubated until mid-log phase (30 °C, 16 h, 230 rpm). Yeast were swapped into 2 L of pABA⁻ media with glucose (0.1%, w/v) and glycerol (3%, w/v) at an initial density of $2.5 \times 10^6$ cells/mL and incubated until early log phase (30 °C, 16 h, 230 rpm). $1 \times 10^8$ cells were collected for whole-cell (WC) analyses. The remaining culture was pelleted by centrifugation ($4,500 \times g$, 7 min) and weighed (2−3 g). Pellets were then fractionated using previously described methods[59]. To isolate crude mitochondria, samples were pelleted by centrifugation ($15,000 \times g$, 10 min, 4 °C) and the supernatant was collected (25−30 mL). Crude mitochondria were resuspended in SEM buffer (10 mM MOPS/KOH pH 7.2, 250 mM sucrose, 1 mM EDTA) containing 10 μg trypsin (sequencing grade, Promega) and rotated end-over-end overnight (16 h, 4 °C) to disrupt proteinaceous organelle contact tethers[60]. Collected supernatant material was then subjected to ultracentrifugation ($106,000 \times g$, 1 h, 4 °C) to pellet microsomes (non-mitochondrial fraction; NM). The post-spin supernatant was immediately removed, and pelleted material was resuspended in 300 μL SEM. On the following day, digested samples were pelleted by centrifugation ($15,000 \times g$, 10 min, 4 °C) and the supernatant was collected. Pelleted material was resuspended in 900 μL SEM buffer containing 1 mM phenylmethylsulfonyl fluoride (SEM+PMSF) to deactivate trypsin. Resuspended material was pelleted ($15,000 \times g$, 10 min, 4 °C) and this was repeated once more. Pelleted crude mitochondria were resuspended in 700 μL SEM +PMSF and then added to a freshly prepared sucrose gradient (bottom to top: 1.5 mL 60% sucrose, 4 mL 32% sucrose, 1.5 mL 23% sucrose, and 1.5 mL 15% sucrose) for separation by ultracentrifugation ($134,000 \times g$, 1 h, 4 °C). Enriched mitochondrial samples were recovered at the 32−60% interface and diluted with 30 mL SEM. Mitochondria were pelleted ($15,000 \times g$, 10 min, 4 °C) and resuspended in fresh SEM (150 μL total). The protein concentration of all subcellular fractions (spheroplasts, SP; crude mitochondria, CM; non-mitochondrial fraction, NM; enriched mitochondria, M) was determined using the Pierce™ BCA Protein Assay Kit (Thermo) before western blot (4 μg) analyses and lipid extractions.

**GFP nanobody**
*Recombinant purification.* pCA528-His-SUMO-GFP nanobody (GFPnb) constructs were transformed into RIPL competent E. coli cells for protein expression. GFPnb was overexpressed in E. coli by autoinduction overnight[61] (37 °C, 4 h; 20 °C, 20 h). Cells were isolated by centrifugation ($4,500 \times g$, 12 min, RT), flash frozen in $N_2$(l) dropwise, and stored at −80 °C. For protein purification, cells were added to a Retsch® mixer mill MM 400 screw-top grinding jar pre-equilibrated with $N_2$(l). The cells were lysed by cryogenic grinding (−196 °C, 30 Hz, 120 s × 3). The ground cell pellet was collected and resuspended end-over-end for 1 h in lysis buffer (160 mM HEPES pH 7.5, 400 mM NaCl, 0.25 mM PMSF, 1 Roche cOmplete™ Protease Inhibitor Cocktail tablet, 500 U Benzonase® Nuclease) at 4 °C. The lysate was clarified by centrifugation ($15,000 \times g$, 30 min, 4 °C). Clarified lysate was added to pre-equilibrated TALON® cobalt resin (Takara Bio) and incubated end-over-end for 1 h at 4 °C. TALON® resin was pelleted by centrifugation ($700 \times g$, 2 min, 4 °C) and washed twice with equilibration buffer (160 mM HEPES pH 7.5, 400 mM NaCl, 0.25 mM PMSF) and twice with wash buffer (160 mM HEPES pH 7.5, 400 mM NaCl, 0.25 mM PMSF, 20 mM imidazole). His-tagged protein was eluted with elution buffer (160 mM HEPES (pH 7.5), 400 mM NaCl, 0.25 mM PMSF, 400 mM imidazole). The eluted protein was concentrated to ~600 μL with an Amicon® Ultra Centrifugal Filter (10 kDa MWCO) and exchanged into equilibration buffer. Concentrated protein elution was centrifuged ($15,000 \times g$, 5 min, 4 °C) to pellet the precipitate and filtered through a 0.22 μM syringe filter. Concentrated protein elution was separated via size exclusion chromatography on a HiLoad™ 16/600 Superdex™ 75 pg. Fractions from the size exclusion chromatography were analyzed by SDS-PAGE, and the fractions containing GFPnb were pooled and concentrated to ~1 mL. The concentration of GFPnb was determined by Bradford assay (Bio-Rad Protein Assay Kit II) and was diluted with equilibration buffer and glycerol to a final concentration of 20 mg/mL protein (160 mM HEPES pH 7.5, 400 mM NaCl, 10% glycerol). The final protein was aliquoted, flash-frozen in $N_2$(l), and stored at −80 °C. Fractions from the protein preparation were analyzed by SDS-PAGE.

*Differential scanning fluorimetry.* The differential scanning fluorimetry method (thermal shift assay) was performed as described previously[62]. Purified recombinant GFPnb was diluted to a final concentration of 4 μM with DSF buffer (100 mM HEPES pH 7.5, 150 mM NaCl) and 1:1250 SYPRO® Orange Dye (Life Tech). Thermal shift data were collected with QuantStudio Real-Time PCR v1.2 software and analyzed with Protein Thermal Shift v1.3 software.

**Native nanodisc isolation**. Individual colonies of S. cerevisiae (BY4741) were used to inoculate synthetic complete media (2% glucose, w/v) starter cultures, which were incubated for 14−16 h (30 °C, 230 rpm). Yeast were diluted to $5 \times 10^6$ cells/ mL in 50 mL pABA⁻ media (2% glucose, w/v) and incubated until mid-log phase

(30 °C, 16 h, 230 rpm). Yeast were swapped into 2 L of pABA⁻ media with glucose (0.1%, w/v) and glycerol (3%, w/v) at an initial density of $2.5 \times 10^6$ cells/mL and incubated until early log phase (30 °C, 16 h, 230 rpm). Yeast cultures were pelleted by centrifugation ($4,500 \times g$, 7 min) and weighed (2−3 g). Pellets were then fractionated using previously described methods[59]. For preparative scale affinity purification, crude mitochondria were resuspended in 50 μL BB7.4 (0.6 M sorbitol, 20 mM HEPES-KOH pH 7.4), diluted in 950 μL ice-cold BB7.S (20 mM HEPES-KOH pH 7.4), vortexed for 10 s (medium setting 8, Vortex Genie), and incubated on ice for 30 min. Swollen mitochondria were then sonicated briefly (1/8″ tip, 20% amplitude) for 2−5 s pulses with 60 s between pulses. Mitoplasts with osmotically ruptured outer membranes were recovered by centrifugation at ($20,000 \times g$, 10 min, 4 °C). After removing the supernatant, each pellet was resuspended with 1 mL of Buffer B (20 mM HEPES-KOH pH 8.0, 200 mM NaCl) containing 2% (w/v) styrene-maleic acid copolymer (SMA, Polyscope SMALP® 25010 P) by repeat pipetting and rotated end-over-end (4 h, 4 °C). Soluble SMA extracts were separated from non-extracted material by centrifugation at $21,000 \times g$ for 10 min at 4 °C. Soluble material was then added to NTA nickel resin (400 μL slurry, Qiagen), which was pre-charged (overnight at 4 °C, end-over-end) with recombinant His-tagged GFPnb (12.5 μL, 20 mg/mL). This mixture of soluble SMA extracts and charged nickel resin was rotated end-over-end (24 h, 4 °C).

Nickel resin was pelleted by centrifugation ($700 \times g$, 2 min, 4 °C) and the supernatant fraction was carefully collected. Nickel resin was washed twice with Buffer B and twice with 500 μL Wash Buffer [Buffer B containing 20 mM imidazole]. Native nanodiscs bound to His-GFPnb were eluted with Buffer B containing 250 mM imidazole by rotating end-over-end for 20 min at 4 °C. Due to the presence of GFP nanobody in the elution samples, relative target abundance was determined by western blot analysis and anti-GFP band quantification. Protein concentrations of all other samples were quantified by Pierce™ BCA Protein Assay Kit (Thermo).

**Lipid extraction**
*$CHCl_3$:MeOH extraction.* $1 \times 10^8$ yeast cells were harvested by centrifugation ($4,000 \times g$, 5 min, 4 °C). The supernatant was removed, and the cell pellet was flash-frozen in $N_2$ (l) and stored at −80 °C. Frozen yeast pellets were thawed on ice and resuspended in 100 μL cold water. To this, 100 μL of glass beads (0.5 mm; RPI) and $CoQ_{10}$ internal standard (10 μL, 10 μM) were added and bead beat (2 min, 4 °C). 900 μL extraction solvent (1:1 $CHCl_3$/MeOH, 4 °C) was added and samples were vortexed briefly. To complete phase separation, samples were acidified with 85 μL 6 M HCl (4 °C), vortexed (2 × 30 s, 4 °C), and centrifuged ($5,000 \times g$, 2 min, 4 °C). The resulting aqueous layer (top) was removed and 400 μL of the organic layer (bottom) was transferred to a clean tube and dried under $Ar_{(g)}$. Dried organic matter (lipids) were reconstituted in $ACN/IPA/H_2O$ (65:30:5, v/v/v, 100 μL) by vortexing (2 × 30 s, RT) and transferred to an amber vial (Sigma; QSertVial™, 12 × 32 mm, 0.3 mL) for LC–MS analysis.

*Petroleum ether:MeOH extraction.* For yeast whole-cell measurements, $1 \times 10^8$ cells were collected by centrifugation ($4,000 \times g$, 5 min) and layered with 100 μL of glass beads (0.5 mm; RPI). Whole-cell samples and all other fractions were then suspended in ice-cold methanol (500 μL; with 1 μM $CoQ_8$ internal standard) and vortexed (10 min, 4 °C). ~500 μL of petroleum ether was added to extract lipids, and samples were vortexed (3 min, 4 °C) and centrifuged ($17,000 \times g$, 1 min) to separate phases. The petroleum ether (upper) layer was collected, and the extraction was repeated with another round of petroleum ether (500 μL), vortexing (3 min, 4 °C), and centrifugation ($17,000 \times g$, 1 min). The petroleum ether layers were pooled and dried under argon. Lipids were resuspended in 2-propanol (15 μL) and transferred to amber glass vials (Sigma; QSertVial™, 12 × 32 mm, 0.3 mL). Sodium borohydride (15 μL of 10 mM in 2-propanol) was added to reduce quinones, and samples were vortexed briefly and incubated (5−10 min). Methanol (20 μL) was added to remove excess sodium borohydride, and samples were vortexed briefly and incubated (5−10 min). Samples were briefly flushed with nitrogen gas.

**Lipidomic analysis**
*Targeted LC-MS for yeast $CoQ_6$ and $PPHB_6$.* LC-MS analysis was performed on an Acquity CSH C18 column held at 50 °C (100 mm × 2.1 mm × 1.7 μm particle size; Waters) using a Vanquish Binary Pump (400 μL/min flow rate; Thermo Scientific). Mobile phase A consisted of 10 mM ammonium acetate and 250 μL/L acetic acid in ACN:H2O (70:30, v/v). Mobile phase B consisted of IPA:ACN (90:10, v/v) also with 10 mM ammonium acetate and 250 μL/L acetic acid. Mobile phase B was initially held at 50% for 1.5 min and then increased to 99% over 7.5 min and held there for 2 min. The column was equilibrated for 2.5 min before the next injection. 10 μL of each extract was injected by a Vanquish Split Sampler HT autosampler (Thermo Scientific) in a randomized order.

The LC system was coupled to a Q Exactive Orbitrap mass spectrometer (MS) through a heated electrospray ionization (HESI II) source (Thermo Scientific). Source conditions were as follows: HESI II and capillary temperature at 350 °C, sheath gas flow rate at 25 units, aux gas flow rate at 15 units, sweep gas flow rate at 5 units, spray voltage at +3.5 kV/−3.5 kV, and S-lens RF at 90.0 units. The MS was operated in a polarity switching mode acquiring positive and negative full MS and

MS2 spectra (Top2) within the same injection. Acquisition parameters for full MS scans in both modes were 17,500 resolution, $1 \times 10^6$ automatic gain control (AGC) target, 100 ms ion accumulation time (max IT), and 200−1600 $m/z$ scan range. MS2 scans in both modes were then performed at 17,500 resolution, $1 \times 10^5$ AGC target, 50 ms max IT, 1.0 $m/z$ isolation window, stepped normalized collision energy (NCE) at 20, 30, 40, and a 10.0 s dynamic exclusion.

Parallel reaction monitoring (PRM) in positive polarity mode was utilized to monitor for two primary adducts, $[M+H]^+$ and $[M+NH_4]^+$, of each CoQ species. For $CoQ_6$, we targeted the mass to charge ratio of 592.449 and 609.475; for $CoQ_8$, 728.574 and 745.601; and for $CoQ_{10}$, 864.7 and 881.727. PRM MS settings were: automatic gain control (AGC) target at $5 \times 10^5$, Maximum IT at 100 ms, resolving power at 35,000, loop count at 2, isolation window at 3.0 $m/z$, and collision energy at 35. Another experiment performed in tandem with PRM used targeted single ion monitoring (t-SIM) in negative mode to determine the primary adduct, $[M-H]^-$, of CoQ intermediates. For $PPHB_6$, we targeted the mass to charge ratio of 544.908 and used the following t-SIM MS settings: AGC target at $5 \times 10^5$, Maximum IT at 100 ms, and resolving power at 140,000 with an isolation window of 4.0 $m/z$.

*Data analysis.* The resulting LC-MS data were manually processed using a custom TraceFinder 4.1 (Thermo Scientific) method using a mass precision of 4 and mass tolerance of 10 ppm to detect and identify the different species and adducts of $CoQ_6$ and $CoQ_8$ and intermediates.

*Targeted HPLC-ECD for yeast $CoQ_6$.* For yeast whole-cell measurements, $5 \times 10^8$ cells were collected by centrifugation (4,000 × $g$, 5 min) and layered with 100 μL of glass beads (0.5 mm; RPI). Lipids from whole-cell samples and other fractions were extracted according to the "*Petroleum Ether:MeOH Extraction*" section above. Samples were analyzed by reverse-phase high-pressure liquid chromatography with electrochemical detection (HPLC-ECD) using a C18 column (Thermo Scientific, Betasil C18, 100 × 2.1 mm, particle size 3 μm) at a flow rate of 0.3 mL/min with a mobile phase of 75% methanol, 20% 2-propanol, and 5% ammonium acetate (1 M, pH 4.4). After separation on the column, the $NaBH_4$-reduced quinones were quantified on ECD detector (Thermo Scientific ECD3000-RS) equipped with 6020RS omni Coulometric Guarding Cell "E1", and 6011RS ultra Analytical Cell "E2" and "E3". To prevent premature quinone oxidation, the E1 guarding electrode was set to −200 mV. Measurements were made using the analytical E2 electrode operating at 600 mV after complete oxidation of the quinone sample and E3 electrode (600 mV) was used to ensure that the total signal was recorded on the E2 cell. For each experiment, a $CoQ_6$ standard in 2-propanol was also prepared with sodium borohydride and methanol treatment, and different volumes were injected to make a standard curve. Quinones were quantified by integrating respective peaks using the Chromeleon 7.2.10 software (Thermo) and normalized to $CoQ_8$ internal standard.

## Antibodies and western blots

*Antibodies.* Primary antibodies used in this study include anti-Kar2 (SCBT sc-33630, 1:5000; RRID: AB_672118), anti-Cit1[63] (custom made at Biomatik, 1:4000), anti-Tom70[64] (1:1000, a gift from Nora Vogtle, University of Freiburg), anti-Vdac (Abcam ab110326, 1:2000; RRID: AB_10865182); anti-GFP (SCBT sc-9996, 1:1000; RRID: AB_627695), anti-Sdh2[65] (1:5000, a gift from Oleh Khalimonchuk, University of Nebraska). Secondary antibodies include goat anti-mouse (LI-COR 926-32210, 1:15000; RRID: AB_621842) and goat anti-rabbit (LI-COR 926-32211, 1:15000; RRID: AB_621843).

*SMA solubility western blot.* Mitoplasts were recovered and solubilized in styrene-maleic acid-containing buffer as described above in "*Native Nanodisc Isolation.*" To determine the extent of GFP target solubilization, equal amounts of "input" (IP) and soluble supernatant (S) were obtained, along with the total pellet (insoluble, IS). Seventy-five microliters of the input sample was collected immediately after SMA solubilization. After separating soluble SMA extracts from non-extracted material via centrifugation (21,000 × $g$, 10 min, 4 °C), the supernatant was transferred to a clean tube for an additional 5-min spin. Seventy-five microliters of the soluble sample was then transferred to a new tube. The resulting pellet was washed with 1 mL of Buffer B and centrifuged (21,000 × $g$ for 5 min at 4 °C). The resulting supernatant was aspirated and 75 μL of Buffer B was added to the insoluble (IS) fraction. From each sample, proteins were extracted by standard chloroform-methanol procedures. Precipitated protein was reconstituted in 75 μl 0.1 M NaOH. 25 μL 4× LDS sample buffer containing beta-mercaptoethanol (BME) was added and samples were incubated (95 °C, ~10 min). Proteins were analyzed with 4–12% Novex NuPAGE Bis-Tris SDS-PAGE (Invitrogen) gels (1 h, 150 V). The gel was transferred to PVDF membrane at 100 V for 1 h with transfer buffer (192 mM glycine, 25 mM Tris, 20% methanol [v/v]). The membrane was blocked with 5% nonfat dry milk (NFDM) in TBST (20 mM Tris pH 7.4, 150 mM NaCl, 0.05% Tween 20 [v/v]) (1 h with agitation). Antibodies were diluted in 1% NFDM in TBST and incubated with the PVDF (overnight, 4 °C with agitation). The PVDF was washed three times in TBST and the secondary antibodies were diluted 1:15,000 in 1% NFDM in TBST (1.5 h, r.t.). The membrane was washed three times in TBST and imaged on a LI-COR Odyssey CLx using Image Studio v5.2 software.

*SMALP fractionation western blot.* Fractions described above in "*Native Nanodisc Isolation*" and "*SMA solubility western Blot*" were collected and used for western blot analysis. Four micrograms of spheroplasts (SP) and crude mitochondria (CM) were loaded, along with equal volumes of extracted soluble (S) and final elution (E) samples. Western blots were performed as described above.

**Statistical analysis.** All experiments were performed in at least biological triplicate, unless stated otherwise. In all cases, "mean" refers to the arithmetic mean, and "SD" refers to sample standard deviation. Statistical analyses were performed using Microsoft Excel. $p$-values were calculated using an unpaired, two-tailed, Student's $t$-test. In all cases, $n$ represents independent replicates of an experiment. For all western blot, Coomassie gel, and drop assay data, a representative blot from three independent experiments is displayed.

**Reporting summary.** Further information on research design is available in the Nature Research Reporting Summary linked to this article.

## Data availability
The next generation sequencing data generated in this study (Fig. 2c, Supplementary Fig. 1c) have been deposited to NCBI SRA (BioProject: accession PRJNA679831). Source data for Figs. 1–4 and Supplementary Figs. 1−4 are provided in the Source Data file. All other data supporting the findings of this study are available from the corresponding authors on reasonable request. Source data are provided with this paper.

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

## Acknowledgements

We thank Steven Claypool and Nathan Alder for consultation and assistance on SMALP generation, Adam Frost for providing the plasmid containing GFP nanobody, Nora Vogtle and Oleh Khalimonchuk for providing mitochondrial antibodies, Jared Rutter and Jodi Nunnari for providing the parental yeast strains used in our studies, Matt Stefely for assistance with figure generation, and current and former members of the Pagliarini Laboratory for their feedback. This work was supported by NIH awards R35 GM131795 and R01 GM112057 (to D.J.P.), P41 GM108538 (to J.J.C. and D.J.P.), and T32 DK007665 (to Z.A.K.); funds from the BJC Investigator Program (to D.J.P.), William H. Peterson Fellowship, Washburn Wharton Fellowship, and University of Wisconsin Biochemistry Funding (to Z.A.K.); and a National Science Foundation Graduate Research Fellowship DGE-1747503 (to K.P.R.).

## Author contributions

Z.A.K., K.P.R. and D.J.P. conceived of the project and its design. Z.A.K. and K.P.R. conducted experiments and performed data analysis. J.M.S. purified and characterized GFP nanobody. M.M. performed and analyzed HPLC-ECD experiments. B.R.P. and P.D.H. performed and analyzed mass spectrometry experiments. J.J.C. oversaw all mass spectrometry-based experiments. A.J. contributed to new reagents (cloning). All authors edited the paper. Z.A.K., K.P.R., and D.J.P. wrote the paper. D.J.P. supervised the project.

## Competing interests

J.J.C. is a consultant for Thermo Fisher Scientific. The authors declare no other competing interests.
