## [Peer Review File · Nature Communications]

REVIEWER COMMENTS

Reviewer #1 (Remarks to the Author):

Exciting work demonstrating the antagonistic role of two paralog proteins in the distribution of coenzymeQ6 (CoQ6) in yeast, albeit lacking mechanistic insight of how this distribution is wrought. The authors identify conditions in which yeast cells rely on CoQ6 abundance in non-mitochondrial membranes to survive a linoleic-acid (oxidative) stress. In a clever screen, they identify mutants that show increased resistance to linoleic acid. One of this mutants is in a gene (they dub CQD1) encoding a paralog of COQ8, hence raising interest. Subcellular fractionation shows that, while these mutants appear to have similar total levels of CoQ6, there is an increase in the non-mitochondrial fraction of it. Further genetics and pharmacological experiments allow to infer a causality: lack of Cqd1 causes an imbalance in CoQ6 distribution, causes accumulation in non-mito membranes, causes linoleate resistance.

The paper then turns to a second paralog Mcp2 (re-named Cqd2). Surprisingly, deletion of CQD2 causes opposite phenotype as that of CQD1 (that is linoleate sensitivity) and double mutations suppress each other phenotypes. Again, Cqd2 mutant cells have normal whole-cell CoQ6 levels. To assess subcellular distribution of CoQ6 in cqd2 mutant cells, the authors turn to a fancy method to extract inner-mito membrane (IMM) nanodiscs by immunopurification. This method reveal an increased and decreased accumulation of CoQ6 in the IMM of cqd2 and cqd1 mutants respectively, and normal levels in double mutants.

Thus, although the mechanism is still unknown, Cqd1 and Cqd2 appear to affect CoQ6 distribution in the cell.

This is a very interesting story despite the lack of mechanistic insight. For most part, the careful combination of genetic epistatic analysis, pharmacological interventions and lipid quantifications support the inferred causalities reasonably well.

There are two points that this reviewer think would be important to truly pin the story down.

1-If CGD1 and 2 affect linoleic acid sensitivity via redistribution of CoQ6, their deletion should have no effect in a CoQ6-deficient mutant. Of course, these mutants are themselves sensitive to linoleic acid, but adapting linoleic acid concentrations should allow to identify conditions where both a positive and a negative contribution of CGD1 or 2 (if any) should be detectable. If no change is observed, then this is a strong epistasis and a good way to exclude the possibility that CGD1/2 affect linoleate sensitivity by other means than CoQ6 distribution.

2-cgd2 mutant cells have similar CoQ6 levels in whole cell membranes but increased levels in IMM. It is therefore inferred that CoQ6 levels must be decreased in non-mitochondrial membranes, causing Linoleic acid sensitivity. It would be better to directly show non-mitochondrial CoQ6 levels in cgd2 mutants, as for cgd1 mutants in figure 3D. This is particularly important since without it, an important point of the paper relies solely on the nanodisc data. These nanodisc data are very exciting and convincing but we do not yet have the necessary hindsight to really understand how the SMALP extraction takes place and whether lipids co-extracted with a protein have not been reshuffled during extraction. It is also odd that the authors chose Sdh2 as IMM marker, as Sdh2 is in complex with Sdh4, the bait of the pull-down. To show that the nanodiscs indeed contains intact IMM, it would have been preferable to blot for an unrelated IMM protein. All of this makes it important to show decreased non-mitochondrial CoQ6 levels in cgd2 mutants using traditional approaches.

Minor points:

Maybe a word about why linoleic acid causes oxidative stress? This is well described in the referenced Do et al. 1996 paper, but could be informative to understand the paper.

CoQ6 is sometimes referred to as CoQ and ubiquinone. Most readers assume ubiquinone to be CoQ10. Maybe a word to explain that CoQ6 is the main form of ubiquinone in yeast?

Reviewer #2 (Remarks to the Author):

The authors describe a novel mechanism whereby two previously uncharacterized mitochondrial proteins, Cqd1 (Ypl109c) and Cqd2 (Ylr253w) reciprocally regulate intracellular coenzyme Q distribution. Their data clearly show that loss of Cqd1 leads to less mitochondrial CoQ, while loss of Cqd2 results in the reverse effect. Overall, the authors present an intriguing phenotype that demonstrates a role for two previously uncharacterized proteins in the subcellular trafficking of CoQ, a topic that is completely unstudied. However, the authors provide little to no mechanistic insight into how Cqd1 and Cqd2 might be performing this role, aside from the requirement for their catalytic activity. While the authors acknowledge this, which is greatly appreciated, the inclusion of additional data to address the mechanism by which Cqd1 and Cqd2 control CoQ distribution is necessary to increase the quality and impact of the paper. With this, it would be a good fit for Nature Communications.

Major comments:

Cqd1 and Cqd2 are tethered to the inner mitochondrial membrane, facing the IMS. As such, it is hard to imagine a direct role for them in extracting CoQ from the mitochondria altogether. However, it is more plausible that Cqd1 and Cqd2 could directly participate in the transport of CoQ between the inner and outer membrane. The authors should test this hypothesis using their membrane patch method and an outer membrane protein such as Porin to quantify the difference between CoQ in the inner and outer membranes.

Similarly, in the discussion, the authors identify a prime candidate to participate in the distribution of CoQ from the OMM to the rest of the cell in the ERMES complex. The authors should test whether ERMES mutants display similar CoQ distribution disruptions as their CQD mutants, and how these phenotypes are related – for example in a *cqd1*, ERMES double mutant, does CoQ accumulate on the OMM, and are these mutants sensitized or resistant to PUFA in their *gpx* triple mutants?

Finally, it is clear from the data that the *mutC* line has other mutations that are consequential for PUFA resistance, in addition to Cqd1, and that the *mutA*, *B*, and *D* lines likely do as well (do they even have a Cqd mutation at all?). The authors should at the very least discuss whether there are other identified mutations in these lines that could potentially correspond to other steps in CoQ trafficking.

Reviewer #3 (Remarks to the Author):

In this study the authors make the very interesting discovery that two yeast orphan mitochondrial proteins function in the trafficking of coenzyme Q (CoQ). Both yeast proteins have human homologues, and are related to the Coq8 atypical kinase, known to be necessary for biosynthesis of CoQ. Intriguingly, the two new proteins, named Cqd1 and Cqd2 (for CoQ distribution), are shown to affect the respective retention or egress of mitochondrial CoQ. The authors used a clever screen to uncover the function of these proteins. The genetic forward screen took advantage of a triple mutant that lacked the phospholipid glutathione peroxidase genes *gpx1*, *gpx2*, and *gpx3*. This triple mutant (termed $\Delta gpx1/2/3$) is known to be sensitive to treatment with the polyunsaturated fatty acid linolenic (18:3). The authors mutagenized the triple mutant and selected for mutants able to grow in the presence of 18:3.

The manuscript is clearly written and the results presented identify two conserved atypical protein kinases that mediate the partitioning of CoQ between mitochondrial and nonmitochondrial membranes. Conserved residues within these kinases are essential to their effects on trafficking. Such subcellular partitioning of CoQ content affects the ability of CoQ to function as an antioxidant in

nonmitochondrial membranes, and as an essential electron transport component within mitochondria.

The paper would benefit if the authors could address the following points:

1. The authors state that extra mitochondrial CoQ is the predominant mediator of PUFA resistance. This is true for the $\Delta gpx1/2,/3$ triple mutant. However, does the triple mutant primarily impact nonmitochondrial antioxidant function? Are any of the Gpx1 – Gpx3 polypeptides located within mitochondrial sites? Is it possible that mitochondria may have a distinct antioxidant protection system? A discussion of the location of the Gpx polypeptides would clarify the author's assertion about nonmitochondrial CoQ being responsible for the CoQ antioxidant activity.
2. Reference 6 (Avery and Avery, 2001) is cited as supporting that exogenous PUFAs are incorporated into endogenous membranes slowly and therefore populate non-mitochondrial membranes first. While this is likely to be true, reference 6 does not provide data that show non-mitochondrial membranes are the first ones populated. In fact, other literature shows that exogenously supplied PUFAs are incorporated into mitochondrial membrane lipids.
3. The authors compare the rescue of PUFA-treated $\Delta gpx1/2,/3$ triple mutant with DecylQ and MitoQ, as a means of claiming that it is the nonmitochondrial location of the DecylQ that bestows antioxidant function. However, unlike DecylQ, MitoQ does not restore respiratory chain activity in yeast mutants lacking CoQ. Thus, it is possible that the participation of DecylQ in redox chemistry within mitochondria, coupled to its fast diffusion may be responsible for the enhanced antioxidant activity as compared to MitoQ.
4. In Figure 3 panel f, the authors show that addition of the diffusible CoQ analogs CoQ2 and CoQ4 restore the growth rate of the $\Delta cqd1$ mutant on growth medium containing a nonfermentable carbon source. It seems reasonable that these small diffusible analogs of CoQ would also rescue the PUFA sensitivity of the $\Delta cqd2$ mutant. It would be intriguing to test the effect of treatment with exogenous CoQ6 on both mutants (the $\Delta cqd1$ and $\Delta cqd2$ mutants). This experiment would suggest that the functions of Cqd1 and Cqd2 might be bypassed by CoQ2 and CoQ4 and could test the idea that Cqd1 and Cqd2 are required to traffic the more hydrophobic CoQ6.
5. It is intriguing that in the absence of both Cqd1 and Cqd2 the $\Delta gpx1/2,/3$ triple mutant regains the resistance to PUFA treatment and growth on medium containing a nonfermentable C source. Does this suggest that in the absence of Cqd1 and Cqd2 there are other trafficking pathways for endogenously synthesized CoQ?
6. It is intriguing that the authors have uncovered additional genes that appear to modulate the antioxidant function of CoQ (e.g. in mutA, mutB and mutD). Perhaps they can be encouraged to discuss the possible functional roles of these candidates.

Minor notes:

Line 460 contains repeated text

Extended Figure 3 panel k, move the GFP label.

Specify whether the linolenic used was alpha or gamma linolenic.

UbiB proteins regulate cellular CoQ distribution
Nature Communications manuscript NCOMMS-20-47913-T

Overall response to referee comments

We thank the reviewers for their careful reading of our work and their helpful comments, which we feel have led to marked improvements in our manuscript. Guided by their critiques, we focused our revision efforts in six primary areas:

1. We have repeated our classical fractionation work with the $\Delta cqd2$ and $\Delta cqd1\Delta cqd2$ strains using an improved protocol. We provide new data that, we believe, strongly support our previous conclusions regarding $\Delta cqd1$ and $\Delta cqd2$. These experiments represent a significant improvement from our previous results, but reveal that this approach is insufficient for capturing loss of non-mitochondrial CoQ in $\Delta cqd2$ samples.
2. We performed additional experiments to establish that our observed PUFA sensitivity phenotypes are indeed CoQ-dependent. We demonstrate that deleting *CQD1* or *CQD2* has no effect in background strains deficient in CoQ.
3. We investigated the CoQ distribution pathway using a genetic approach to examine the involvement of several mitochondrial proteins and pathways. We disrupted important protein tethers in ERMES (*MDM34*), MICOS (*MIC60*), vCLAMP (*MCP1*), as well as two additional genes previously associated with mitochondrial lipid homeostasis—*LTC1* and *MDM31*. Remarkably, using our PUFA sensitivity assay, we found that none of these genes were essential for CoQ export from mitochondria.
4. We probed three additional mutated genes from our forward-genetic screen (*FAA1*, *NDE2*, *DNF3*) for altered PUFA resistance, but found no significant changes compared to $\Delta gpx1/2/3$ yeast. Although these genes do not appear to be connected to this phenotype, we contend that these results further support the significance of our original *CQD1* discovery.
5. We have attempted an OMM SMALP purification using Tom20-GFP yeast, but—predictably given the nascent stages of this technique—were unsuccessful in sufficiently removing the more prevalent IMM content, preventing a conclusive analysis of an OMM microenvironment.
6. We have included notable additions to the text regarding PUFA toxicity, CoQ nomenclature, Cqd1/2 mechanistic partners, and yeast screen causative mutations.

Point-by-point response to referee comments:

Reviewer #1 (Remarks to the Author):

Exciting work demonstrating the antagonistic role of two paralog proteins in the distribution of coenzymeQ6 (CoQ6) in yeast, albeit lacking mechanistic insight of how this distribution is wrought. The authors identify conditions in which yeast cells rely on CoQ6 abundance in non-

mitochondrial membranes to survive a linoleic-acid (oxidative) stress. In a clever screen, they identify mutants that show increased resistance to linoleic acid. One of these mutants is in a gene (they dub CQD1) encoding a paralog of COQ8, hence raising interest. Subcellular fractionation shows that, while these mutants appear to have similar total levels of CoQ6, there is an increase in the non-mitochondrial fraction of it. Further genetics and pharmacological experiments allow to infer a causality:

1. Lack of Cqd1 causes an imbalance in CoQ6 distribution,
2. Causes accumulation in non-mito membranes,
3. Causes linoleate resistance.

The paper then turns to a second paralog Mcp2 (re-named Cqd2). Surprisingly, deletion of CQD2 causes opposite phenotype as that of CQD1 (that is linoleate sensitivity) and double mutations suppress each other phenotypes. Again, Cqd2 mutant cells have normal whole-cell CoQ6 levels. To assess subcellular distribution of CoQ6 in cqd2 mutant cells, the authors turn to a fancy method to extract inner-mito membrane (IMM) nanodiscs by immunopurification. This method reveals an increased and decreased accumulation of CoQ6 in the IMM of cqd2 and cqd1 mutants respectively, and normal levels in double mutants. Thus, although the mechanism is still unknown, Cqd1 and Cqd2 appear to affect CoQ6 distribution in the cell.

This is a very interesting story despite the lack of mechanistic insight. For most part, the careful combination of genetic epistatic analysis, pharmacological interventions and lipid quantifications support the inferred causalities reasonably well.

We thank the reviewer for the overall positive review of our work.

There are two points that this reviewer thinks would be important to truly pin the story down.

1) If CQD1 and 2 affect linoleic acid sensitivity via redistribution of CoQ6, their deletion should have no effect in a CoQ6-deficient mutant. Of course, these mutants are themselves sensitive to linoleic acid, but adapting linoleic acid concentrations should allow to identify conditions where both a positive and a negative contribution of CQD1 or 2 (if any) should be detectable. If no change is observed, then this is a strong epistasis and a good way to exclude the possibility that CQD1/2 affect linoleate sensitivity by other means than CoQ6 distribution.

We thank the reviewer for this suggestion and agree with their reasoning. To address this point, we performed an experiment to assess the impact of CQD1 or CQD2 deletion in CoQ-deficient yeast. We constructed yeast strains unable to synthesize CoQ by deleting COQ2, an essential prenyltransferase in CoQ biosynthesis. Our results show that in $\Delta gpx1/2/3\Delta coq2$ (Rebuttal Fig. 1) or $\Delta coq2$ yeast, subsequent deletion of CQD1 or CQD2 indeed has no impact on linolenic acid

Rebuttal Fig. 1: CQD1 activity is dependent on CoQ.

sensitivity, providing further support of a CoQ-dependent phenotype. This data has been added as Extended Data Fig. 1d-e, and we inserted text referencing these panels on lines 95-97.

2) *cqd2* mutant cells have similar CoQ6 levels in whole cell membranes but increased levels in IMM. It is therefore inferred that CoQ6 levels must be decreased in non-mitochondrial membranes, causing Linoleic acid sensitivity. It would be better to directly show non-mitochondrial CoQ6 levels in *cqd2* mutants, as for *cgd1* mutants in figure 3D. This is particularly important since without it, an important point of the paper relies solely on the nanodisc data. These nanodisc data are very exciting and convincing but we do not yet have the necessary hindsight to really understand how the SMALP extraction takes place and whether lipids co-extracted with a protein have not been reshuffled during extraction. All of this makes it important to show decreased non-mitochondrial CoQ6 levels in *cgd2* mutants using traditional approaches.

Rebuttal Fig. 2: Cellular distribution of CoQ in CQD deletion strains. Cells lacking *CQD1* or *CQD2* have decreased and increased levels of mitochondrial CoQ, respectively, and the double deletion possess an intermediate phenotype. The concomitant increase in non-mitochondrial (NM) CoQ is seen in the $\Delta cqd1$ strain, but loss of NM in the $\Delta cqd2$ strain is not detectable by this method.

clearly demonstrate the $\Delta cqd1$ CoQ distribution phenotype. We performed a similar experiment including the $\Delta cqd2$ and $\Delta cqd1\Delta cqd2$ strains, and we have added these results as Extended Data Fig. 3f and Fig. 4e (Rebuttal Fig. 2).

First, we trust the reviewer will agree that these results unambiguously demonstrate the significant impact of *CQD1* deletion, which reproducibly shows large changes in both mitochondrial and non-mitochondrial fractions. Second, similar to what we observed in our SMALP results (now Fig. 4h), we believe our improved method clearly shows that deletion of *CQD2* increases mitochondrial CoQ *without altering whole cell CoQ levels* (see Fig. 4e). Thus, in $\Delta cqd2$ yeast, the elevated CoQ must come at the expense of CoQ levels elsewhere in the cell. However, we were not able to capture this non-mitochondrial CoQ depletion for at least two reasons. First, our method is much better equipped to detect a gain-of-signal (e.g., the $\Delta cqd1$ phenotype) because obtaining extra-pure mitochondria results in very low yields of other fractions (i.e., it is very difficult to obtain large amounts of a cellular component without some level of mitochondrial contamination, which would swamp the signal due to the high CoQ concentration

We agree that this is a fair and logical request, and we spent a significant amount of time and effort during our revision performing these experiments. To begin, we repeated the classical fractionation approach to measure CoQ from mitochondrial and non-mitochondrial fractions, and, in an effort to improve our separation, sample yield was sacrificed to ensure maximum purity. We are excited to report that we were successful in improving sample purity, as shown in our new Extended Data Fig. 2a. We have also updated Fig. 3d with the new CoQ abundance results from this improved method, which now more

in mitochondria). For this reason, both $\Delta cqd2$ and WT yeast have a very low CoQ signal in the “NM” fraction (see above). Second, CoQ derived from mitochondria becomes incorporated in all cellular membranes, but our method only captures some fraction of those membranes. That is, if ten units of CoQ were exported from mitochondria, they might become incorporated into ten separate cellular destinations, yet our method might only capture a few. Solving this issue would require significantly scaling up our method (100s of liters of culture) and establishing the parameters to obtain a highly pure and more inclusive non-mitochondrial fractions lacking any mitochondrial contamination. We estimate that this effort would require multiple additional months and may still fail to have the sensitivity and precision to capture the expected ~two-fold loss of CoQ in the $\Delta cqd2$ NM fraction.

Overall, we hope the reviewer agrees that the preponderance of evidence (elevated mitochondrial CoQ, insignificant changes to whole cell CoQ, the respiratory phenotype, SMALP data, intermediate double deletion phenotype, etc.) clearly points toward a shift in CoQ distribution. Nonetheless, since we cannot definitively demonstrate decreased levels of CoQ in the $\Delta cqd2$ non-mitochondrial samples as requested, we have updated our Fig. 4 model (now Fig. 4i) to reflect the uncertainty of this particular aspect.

It is also odd that the authors chose Sdh2 as IMM marker, as Sdh2 is in complex with Sdh4, the bait of the pull-down. To show that the nanodiscs indeed contain intact IMM, it would have been preferable to blot for an unrelated IMM protein.

We appreciate the opportunity to clarify this point in our manuscript. Our original intent of blotting for α -Sdh2 was not as an IMM marker, but rather as a known and soluble Complex II subunit that would be expected to co-purify with Sdh4-GFP IMM patches. Published work suggests SMALP particles are ~10 nm in diameter (Dörr et al., *Eur Biophys J*, 2016), which makes the possibility of multiple IMM proteins existing in the same patch unlikely. Regardless, we contend that the presence of another IMM protein would neither support nor challenge our previous conclusions.

Minor points:

Maybe a word about why linoleic acid causes oxidative stress? This is well described in the referenced Do et al. 1996 paper, but could be informative to understand the paper.

We thank the reviewer for this suggestion and have added supporting text on lines 41-43.

CoQ6 is sometimes referred to as CoQ and ubiquinone. Most readers assume ubiquinone to be CoQ10. Maybe a word to explain that CoQ6 is the main form of ubiquinone in yeast?

We thank the reviewer for this suggestion. To ensure clarity, we updated text on lines 37-38.

Reviewer #2 (Remarks to the Author):

The authors describe a novel mechanism whereby two previously uncharacterized mitochondrial proteins, Cqd1 (Ypl109c) and Cqd2 (Ylr253w) reciprocally regulate intracellular

coenzyme Q distribution. Their data clearly show that loss of *Cqd1* leads to less mitochondrial CoQ, while loss of *Cqd2* results in the reverse effect. Overall, the authors present an intriguing phenotype that demonstrates a role for two previously uncharacterized proteins in the subcellular trafficking of CoQ, a topic that is completely unstudied. However, the authors provide little to no mechanistic insight into how *Cqd1* and *Cqd2* might be performing this role, aside from the requirement for their catalytic activity. While the authors acknowledge this, which is greatly appreciated, the inclusion of additional data to address the mechanism by which *Cqd1* and *Cqd2* control CoQ distribution is necessary to increase the quality and impact of the paper. With this, it would be a good fit for *Nature Communications*.

We thank the reviewer for their review of our work. Indeed, the recalcitrance of these proteins to purification has been, and continues to be, a roadblock to deeper mechanistic investigation. Nonetheless, to begin addressing this point, we have updated our manuscript with new analyses described below.

Major comments:

Cqd1 and *Cqd2* are tethered to the inner mitochondrial membrane, facing the IMS. As such, it is hard to imagine a direct role for them in extracting CoQ from the mitochondria altogether. However, it is more plausible that *Cqd1* and *Cqd2* could directly participate in the transport of CoQ between the inner and outer membrane. The authors should test this hypothesis using their membrane patch method and an outer membrane protein such as Porin to quantify the difference between CoQ in the inner and outer membranes.

We agree with the reviewer's analysis and hypothesis concerning CoQ content in the OMM. That said, isolation of an OMM SMALP has not yet been demonstrated in the literature, and classical methods are sufficient only for obtaining an enriched (i.e., not pure) OMM sample, and only when using growth conditions that are incompatible with our experiment given the demonstrated $\Delta cqd1$ phenotype (Vogtle et al., *Nat Comms*, 2017; Morgenstern et al., *Cell Rep*, 2017). It is unclear how useful a merely enriched sample would be, as IMM contamination would be expected to contribute significant (and inconsistent) amounts of CoQ to our sample analysis, confounding data interpretation.

Therefore, we attempted an OMM SMALP isolation akin to our previous results, as suggested. For this work, we used an endogenously tagged Tom20-GFP strain from the same BY4741 GFP-tagged collection (Huh et al., *Nature*, 2003). However, Western blot analyses of our

Rebuttal Fig. 3: Attempted OMM SMALP purification.

Western blot analysis of OMM SMALP samples collected from a panel of Tom20-GFP yeast strains. α -Cox2 and α -Sdh2 antibodies were used to assess the extent of IMM contamination in the elution samples.

elution samples clearly indicated the presence of IMM contamination, based on α -Sdh2 and α -Cox2 blotting (Rebuttal Fig. 3 above). We note that similar work is ongoing in our collaborators' laboratories, which, after multiple years, has also not yielded pure OMM SMALPs (Steven Claypool, personal communication). We and others are continuing this work, but we hope the reviewer will agree that the work required to obtain a pure OMM by the still nascent SMALP methodology is, unfortunately beyond the scope and time frame of this manuscript. Nonetheless, we agree with the reviewer's model and interpretation of our data.

While this does not directly address the reviewer's specific point, we note that we have invested significant effort during our revision to improving our overall mitochondrial purification method, which has resulted in cleaner and more quantifiable results for the CoQ distribution panels in Figures 3 and 4.

*Similarly, in the discussion, the authors identify a prime candidate to participate in the distribution of CoQ from the OMM to the rest of the cell in the ERMES complex. The authors should test whether ERMES mutants display similar CoQ distribution disruptions as their CQD mutants, and how these phenotypes are related – for example in a *cqd1*, ERMES double mutant, does CoQ accumulate on the OMM, and are these mutants sensitized or resistant to PUFA in their *gpx* triple mutants?*

We thank the reviewer for making this suggestion and agree that exploring roles for candidates like ERMES is an important next step in probing the mechanisms of cellular CoQ distribution. At the reviewer's suggestion, we tested the requirement of ERMES in CoQ distribution from mitochondria by deleting *MDM34*, a mitochondrial ERMES subunit, from our $\Delta gpx1/2/3$ and $\Delta gpx1/2/3\Delta cqd1$ strains. As noted by the reviewer, deletion of an essential trafficking bridge may impact PUFA resistance alone or in combination with *CQD1* deletion.

We hypothesized that if ERMES is required for CoQ exit from mitochondria, deletion of *MDM34* would ablate the enhanced PUFA resistance observed in the $\Delta cqd1$ strain. However, we

Rebuttal Fig. 4: Specific mitochondrial contact site subunits and lipid homeostasis genes are not required for increased PUFA resistance in $\Delta cqd1$ strains. See Extended Data Fig. 4 for further details.

found that deleting *MDM34* had no impact on *CQD1* deletion-induced PUFA resistance, suggesting that ERMES is not required for CoQ export from mitochondria (Rebuttal Fig. 4a). A direct comparison of *MDM34* and *CQD1* deletions was not possible since $\Delta gpx1/2/3$ and $\Delta gpx1/2/3\Delta mdm34$ yeast grow at very different rates, but we

indirectly observe that $\Delta gpx1/2/3\Delta mdm34$ yeast are more sensitive to PUFA than $\Delta gpx1/2/3$ yeast. This effect and the difference in growth rates is likely not surprising given the profound effects on yeast growth and mitochondrial morphology caused by deletion of ERMES subunits (Kornmann et al., *Science*, 2009).

To explore this approach more thoroughly, we next deleted subunits of MICOS (*MIC60*) and vCLAMP (*MCP1*), as well as genes associated with intramitochondrial (*MDM31*) (Tamura et al., *J Biol Chem*, 2012; Osman et al., *J Cell Biol*, 2009) and interoganellar (*LTC1*) (Murley et al., *J Cell Biol*, 2015) lipid homeostasis, in both $\Delta gpx1/2/3$ and $\Delta gpx1/2/3\Delta cqd1$ strains. Remarkably, none of these deletions blocked the increased PUFA resistance upon *CQD1* knockout (Rebuttal Fig. 4b-d), suggesting that none of these genes are required for mitochondrial CoQ export. Despite the lack of functional interactions, we view these as important experiments for determining mechanistic partners of *Cqd1* moving forward. We have added these results as Extended Data Fig. 4, and we have inserted corresponding text at the end of the results section (lines 222-228).

Finally, it is clear from the data that the mutC line has other mutations that are consequential for PUFA resistance, in addition to Cqd1, and that the mutA, B, and D lines likely do as well (do they even have a Cqd mutation at all?). The authors should at the very least discuss whether there are other identified mutations in these lines that could potentially correspond to other steps in CoQ trafficking.

We appreciate the opportunity to comment on this important point. Each resistant strain (mutA-D) has over 100 protein-coding mutations (which is typical for EMS-based yeast genetic screens), and we agree that there is likely a combination of genetic perturbations contributing to PUFA resistance. We have updated the text (lines 88-90) to make this more evident.

Rebuttal Fig. 5: Analysis of other candidates from our PUFA sensitivity screen. PUFA resistance of three gene hits from our screen, *NDE2*, *FAA1*, and *DNF3* were deleted from $\Delta gpx1/2/3$ and investigated for changes in PUFA resistance (similar to *CQD1* deletion).

we have updated text in the discussion to express the point (lines 262-263). We look forward to continuing this line of investigation moving forward, perhaps including newer screening platforms (e.g., SATAY).

To begin exploring the potential role(s) of other hits from our screen, we deleted three additional genes (Rebuttal Fig. 5); however, none of these deletions increased PUFA resistance in the $\Delta gpx1/2/3$ strain. We agree with the reviewer that because other strains lacked an apparent *CQD1* mutation, we will ultimately be able to find other candidates using a more comprehensive panel of hit gene deletions in the future, and

Reviewer #3 (Remarks to the Author):

*In this study the authors make the very interesting discovery that two yeast orphan mitochondrial proteins function in the trafficking of coenzyme Q (CoQ). Both yeast proteins have human homologues, and are related to the Coq8 atypical kinase, known to be necessary for biosynthesis of CoQ. Intriguingly, the two new proteins, named Cqd1 and Cqd2 (for CoQ distribution), are shown to affect the respective retention or egress of mitochondrial CoQ. The authors used a clever screen to uncover the function of these proteins. The genetic forward screen took advantage of a triple mutant that lacked the phospholipid glutathione peroxidase genes *gpx1*, *gpx2*, and *gpx3*. This triple mutant (termed Δ *gpx1/2/3*) is known to be sensitive to treatment with the polyunsaturated fatty acid linolenic (18:3). The authors mutagenized the triple mutant and selected for mutants able to grow in the presence of 18:3.*

The manuscript is clearly written and the results presented identify two conserved atypical protein kinases that mediate the partitioning of CoQ between mitochondrial and nonmitochondrial membranes. Conserved residues within these kinases are essential to their effects on trafficking. Such subcellular partitioning of CoQ content affects the ability of CoQ to function as an antioxidant in nonmitochondrial membranes, and as an essential electron transport component within mitochondria.

We thank the reviewer for their overall positive review of our work.

The paper would benefit if the authors could address the following points:

*1. The authors state that extra mitochondrial CoQ is the predominant mediator of PUFA resistance. This is true for the Δ *gpx1/2/3* triple mutant. However, does the triple mutant primarily impact nonmitochondrial antioxidant function? Are any of the *Gpx1* – *Gpx3* polypeptides located within mitochondrial sites? Is it possible that mitochondria may have a distinct antioxidant protection system? A discussion of the location of the *Gpx* polypeptides would clarify the author's assertion about nonmitochondrial CoQ being responsible for the CoQ antioxidant activity.*

SGD provides two resources for the subcellular localization of the *Gpx* enzymes: 1) The localization and quantitation atlas of the yeast proteome (LoQAtE, from the Weizmann Institute), and 2) TheCellVision resource including the CYCLOPs localization database. Both of these resources annotate all three *Gpx* enzymes as being primarily cytosolic. Nonetheless, prompted by the reviewer's question, we investigated additional sources. Other recent sub-mitochondrial profiling efforts (Vogle et al., *Nat Comms*, 2017; Morgenstern et al., *Cell Rep*, 2017) largely conclude the same; however, these and work from Ukai et al., *Biochem Biophys Res Commun*, 2011 also suggest that *Gpx1* and *Gpx3* may have peripheral/conditional mitochondrial interactions, and that *Gpx2* may have a more substantial mitochondrial presence. Therefore, while the preponderance of evidence supports extramitochondrial localizations for these enzymes, it is certainly likely that *Gpx1-3* could also contribute to antioxidant protection in mitochondria. If so, then our screen may also be equipped to identify genes that protect against loss of mitochondrial-

based Gpx defenses. We now note this point in our discussion (lines 265-267). Nonetheless, given that the loss of *CQD1* confers PUFA protection concomitant with a reduction in mitochondrial CoQ, we trust that the reviewer will agree that our screen, as designed, is capable of detecting genes that aid in PUFA resistance via CoQ redistribution.

2. Reference 6 (Avery and Avery, 2001) is cited as supporting that exogenous PUFAs are incorporated into endogenous membranes slowly and therefore populate non-mitochondrial membranes first. While this is likely to be true, reference 6 does not provide data that show non-mitochondrial membranes are the first ones populated. In fact, other literature shows that exogenously supplied PUFAs are incorporated into mitochondrial membrane lipids.

We appreciate the reviewer's point here and have modified our text appropriately, which now states, "This is consistent with previous results showing that exogenous PUFAs are incorporated into endogenous membranes slowly (6) and, therefore, likely populate non-mitochondrial membranes first." Given the results in Fig. 5A from this reference, we still feel that it is likely that exogenous PUFAs will first integrate into the plasma membrane and perhaps organelles more associated with the cell periphery, likely giving them an outsized importance for cell viability in our system. This is supported by our result that MitoQ antioxidant rescue is 10-fold less effective than DecylQ despite its rapid accumulation and elevated effective concentration in mitochondria (Kelso et al., *J Biol Chem*, 2001). Nonetheless, similar to the point on Gpx2 above, the fact that PUFAs do likely become incorporated into mitochondrial membranes means that our screen may also be capable of detecting genes whose deletion helps resist loss of Gpx defenses at/within mitochondria.

3. The authors compare the rescue of PUFA-treated Δ gpx1/2/3 triple mutant with DecylQ and MitoQ, as a means of claiming that it is the nonmitochondrial location of the DecylQ that bestows antioxidant function. However, unlike DecylQ, MitoQ does not restore respiratory chain activity in yeast mutants lacking CoQ. Thus, it is possible that the participation of DecylQ in redox chemistry within mitochondria, coupled to its fast diffusion may be responsible for the enhanced antioxidant activity as compared to MitoQ.

We agree with the reviewer that DecylQ indeed has the ability to participate in the respiratory chain activity, whereas MitoQ seems not to. However, we would like to remind the reviewer that these rescue experiments were conducted under fermentative conditions where mitochondrial respiration is not required. Given this, we believe the reviewer will agree that a mode-of-action by which DecylQ is protective of PUFA-induced oxidative stress due to respiratory chain participation does not seem to follow. Instead, loss of the Gpx enzymes clearly compromises cellular antioxidant defense. Both DecylQ and MitoQ have strong antioxidant activity. Thus, while we cannot fully rule out respiratory chain activity as a contributing factor, our analysis relies on the antioxidant activity and relative localization of these two molecules rather than their respiratory chain functions.

4. In Figure 3 panel f, the authors show that addition of the diffusible CoQ analogs CoQ2 and CoQ4 restore the growth rate of the Δ cqd1 mutant on growth medium containing a

nonfermentable carbon source. It seems reasonable that these small diffusible analogs of CoQ would also rescue the PUFA sensitivity of the $\Delta cqd2$ mutant. It would be intriguing to test the effect of treatment with exogenous CoQ6 on both mutants (the $\Delta cqd1$ and $\Delta cqd2$ mutants). This experiment would suggest that the functions of Cqd1 and Cqd2 might be bypassed by CoQ2 and CoQ4 and could test the idea that Cqd1 and Cqd2 are required to traffic the more hydrophobic CoQ6.

We agree with the reviewer that CoQ analogs should rescue $\Delta cqd2$ PUFA sensitivity. We have performed this experiment with CoQ₂, and indeed, supplementation with CoQ₂ rescues $\Delta gpx1/2/3$, $\Delta gpx1/2/3\Delta cqd1$, and $\Delta gpx1/2/3\Delta cqd2$ strains (Rebuttal Fig. 6). As suggested, we also performed

Rebuttal Fig. 6: CoQ analog rescue of $\Delta cqd1$ and $\Delta cqd2$ PUFA sensitivity.

experiments with CoQ₆ supplementation, but there is no significant difference upon CoQ₆ addition for any of these strains. Based on previous work in our laboratory, we interpreted this as a solubility/availability issue. In our experience, it can take several days for CoQ₆ to rescue a respiratory deficiency, and this result is consistent with our previous observations. Regardless, in both cases, these experiments are assessing the ability of exogenous CoQ to alleviate PUFA stress, whereas our findings deal with redistribution of CoQ from mitochondria, where Cqd1 and Cqd2 reside.

5. It is intriguing that in the absence of both Cqd1 and Cqd2 the $\Delta gpx1/2/3$ triple mutant regains the resistance to PUFA treatment and growth on medium containing a nonfermentable C source. Does this suggest that in the absence of Cqd1 and Cqd2 there are other trafficking pathways for endogenously synthesized CoQ?

This is an important point, and we agree with the reviewer's conclusion. We have updated text in the discussion to more clearly make this point. We also note that this is consistent with other aspects of mitochondrial lipid trafficking, which is reported to be highly redundant (Tamura et al., *J Biochem*, 2019). Importantly, our identification of Cqd1 and Cqd2 should accelerate discovery of other proteins in these pathways (e.g., by performing similar screens in a $\Delta cqd1\Delta cqd2$ background.)

6. It is intriguing that the authors have uncovered additional genes that appear to modulate the antioxidant function of CoQ (e.g., in *mutA*, *mutB* and *mutD*). Perhaps they can be encouraged to discuss the possible functional roles of these candidates.

Indeed, our screen almost certainly includes other gene hits that should prove relevant to these pathways, especially since mutA, mutB, and mutD do not include *CQD1* mutations. However, we believe that the sheer number of possibilities from this EMS-based screen makes it premature to engage in broad speculation at this point. Nonetheless, in an attempt to begin exploring this point (see also our response to Reviewer #2), we have updated the text and tested three additional hits from our screen, but none of these seem to affect PUFA sensitivity (Rebuttal Fig. 5). We look forward to continuing this line of investigation moving forward, perhaps including newer and more comprehensive screening platforms (e.g., SATAY).

Minor notes:

Line 460 contains repeated text

Extended Figure 3 panel k, move the GFP label.

Specify whether the linolenic used was alpha or gamma linolenic.

We thank the reviewer for identifying these errors. Text in line 460 (now line 506) has been removed, the label has been moved from former Extended Data Fig. 3k (now 3l), and we have specified that alpha linolenic acid was used throughout our work.

REVIEWERS' COMMENTS

Reviewer #1 (Remarks to the Author):

The authors did a great job at addressing the reviewer's points. Their assessment of epistasis between CoQ6 biosynthesis mutants and CQD mutants and their assessment of CoQ6 accumulation in non-mitochondrial membranes (despite its scarcity) are entirely convincing.

I have no other points.

Reviewer #2 (Remarks to the Author):

While most of the experiments that were done to address the concerns raised by this reviewer were negative and didn't lead to much additional insight, they authors did the appropriate experiments. So, the situation is not ideal as the mechanistic insight is still not substantial, but the paper is probably appropriate for Nature Communications at this point.

Reviewer #3 (Remarks to the Author):

The authors have done a commendable job responding to the comments of the reviewers. As the authors state, the manuscript has been strengthened further by the added experiments and text.

The only comment I have regards the experiment that tested the effect of the deletion of the MDM34 ERMES subunit gene on the CoQ transport to non-mitochondrial membranes in the *cqd1* mutant. Since previous work suggested that the ERMES mutants have increased levels of non-mitochondrial CoQ (ref #28), the results of this experiment with the *cqd1 mdm34* double mutant were not particularly surprising.

UbiB proteins regulate cellular CoQ distribution in *Saccharomyces cerevisiae*
Nature Communications manuscript NCOMMS-20-47913-T

Overall response to referee comments

We thank the reviewers and editors for their careful reading of our work. Guided by their critiques, we have provided additional revisions to comply with formatting and organizational expectations set forth by Nature Communications.

Point-by-point response to referee comments:

Reviewer #1 (Remarks to the Author):

The authors did a great job at addressing the reviewer's points.

Their assessment of epistasis between CoQ6 biosynthesis mutants and CQD mutants and their assessment of CoQ6 accumulation in non-mitochondrial membranes (despite its scarcity) are entirely convincing.

I have no other points.

We thank the reviewer for the positive review of our revised manuscript.

Reviewer #2 (Remarks to the Author):

While most of the experiments that were done to address the concerns raised by this reviewer were negative and didn't lead to much additional insight, they authors did the appropriate experiments. So, the situation is not ideal as the mechanistic insight is still not substantial, but the paper is probably appropriate for Nature Communications at this point.

We thank the reviewer for the overall positive review of our revised manuscript.

Reviewer #3 (Remarks to the Author):

The authors have done a commendable job responding to the comments of the reviewers. As the authors state, the manuscript has been strengthened further by the added experiments and text.

*The only comment I have regards the experiment that tested the effect of the deletion of the MDM34 ERMES subunit gene on the CoQ transport to non-mitochondrial membranes in the *cqd1* mutant. Since previous work suggested that the ERMES mutants have increased levels of non-mitochondrial CoQ (ref #28), the results of this experiment with the *cqd1 mdm34* double mutant were not particularly surprising.*

We thank the reviewer for the positive review of our revised manuscript. We appreciate their comment regarding the *MDM34* deletion experiments, and we hope that our experimental approach helps to further inform future investigations on this topic.